# Language Model Circuits Are Sparse in the Neuron Basis

**Aryaman Arora** [* 1 2]  **Zhengxuan Wu** [* 1 2]  **Jacob Steinhardt** [1]  **Sarah Schwettmann** [1]

## Abstract

The high-level concepts that a neural network uses to perform computation need not be aligned to individual neurons (Smolensky, 1986). Language model interpretability research has thus turned to techniques which decompose the neuron basis into more interpretable units of model computation, such as sparse autoencoders (SAEs). However, not all neuron-based representations are uninterpretable. For the first time, we empirically show that **MLP neurons are as sparse a feature basis as SAEs**. We use this finding to develop an end-to-end gradient-based attribution pipeline for circuit tracing on the MLP neuron basis, which surfaces causally effective neurons on a variety of tasks. On a standard subject-verb agreement benchmark (Marks et al., 2025), a circuit of $\approx 10^2$ MLP neurons is enough to control model behaviour. On the multi-hop city-state-capital task from Lindsey et al. (2025), we find a circuit in which small sets of neurons encode specific latent reasoning steps (e.g. mapping a city to its state), and can be steered to change the model's output. This work thus advances automated interpretability of language models without imposing additional training costs.

 github.com/TransluceAI/circuits

## 1. Introduction

To help oversee AI systems and ensure their safety, we seek to scalably and faithfully understand their internal computations (Ngo et al., 2022; Cotra, 2022). One approach towards this goal is **circuit tracing**, which aims to localise particular behaviors to interactions between subcomponents of the model (Olah et al., 2020; Elhage et al., 2021; Wang et al., 2023; Conmy et al., 2023; Marks et al., 2025; Ameisen et al.,

2025). Faithful circuits may reveal internal reasoning steps which are not verbalised by the model, enabling us to better oversee systems both at inference time and over the course of training (Sharkey et al., 2025; Casper et al., 2024); beyond oversight, access to circuits can help to control model behaviour via steering (Li et al., 2023; Marks et al., 2025).

To construct circuits, we need discrete units of analysis (*features*) over which to trace computations. It is generally believed that the neurons themselves do not comprise an interpretable base for analysis and features may be distributed across neurons (e.g. Cunningham et al., 2023), which has led to many techniques for creating alternative feature bases that are aspirationally interpretable, usually by sparsifying the original representations. These include sparse dictionary learning (SAEs; Marks et al., 2025), transcoders (Dunefsky et al., 2024; Ge et al., 2024), and CLTs (Ameisen et al., 2025), as well as directly training the model to have sparse circuits (Gao et al., 2025b).

In this work, we question the belief that neuron circuits are not sparse. Specifically, we show that **we can find circuits in the neuron basis that are equally sparse and faithful as those obtained from SAE features**. We propose two changes which close the gap between neurons and SAEs:

1. Prior work uses the post-down projection MLP *outputs* as a baseline comparison to learned features; we instead use the pre-down projection MLP *activations*, which are a privileged basis and yield significantly sparser circuits.

2. Most circuit tracing work uses Integrated Gradients (Sundararajan et al., 2017), which has been found to be noisy on deep models and expensive to compute accurately (Makino et al., 2024). When we switch to a stronger and more efficient attribution method (RelP; Jafari et al., 2025) we close the remaining gap.

Furthermore, **neuron bases are more faithful**. Learned bases are an imperfect approximation of the original model, leading to uninterpretable error terms (Engels et al., 2025; Gurnee, 2024); learned features suffer from splitting and absorption, leading to polysemanticity (Bricken et al., 2023; Chanin et al., 2025); and learned bases are difficult to apply them throughout a model's training process without additional computational overhead (Minder et al., 2025). These

---

[1]Transluce [2]Stanford University. Correspondence to: Aryaman Arora <aryaman@transluce.org>, Zhengxuan Wu <zen@transluce.org>.

*Proceedings of the $43^{rd}$ International Conference on Machine Learning*, Seoul, South Korea. PMLR 306, 2026. Copyright 2026 by the author(s).

drawbacks makes it desirable to use the original neuron basis.

To showcase the utility of our methods, we replicate a case study from recent work on cross-layer transcoders (Lindsey et al., 2025), obtaining analogous results in the neuron basis of Llama 3.1-8B-Instruct.

## 2. Related work

**Causal interpretability.** Language model interpretability uses interventions on internals and measures the resulting effects on outputs in order to establish causal relationships between internals and observed behaviours (Giulianelli et al., 2018; Vig et al., 2020; Geiger et al., 2021; Meng et al., 2022; Chan et al., 2022; Wang et al., 2023; Goldowsky-Dill et al., 2023; Guerner et al., 2023; Geiger et al., 2025). Interventions have enabled understanding how concepts are encoded in internal representations (Geiger et al., 2024), including linguistic features (Lasri et al., 2022; Hanna et al., 2023; Arora et al., 2024), mathematical reasoning (Wu et al., 2023; Baeumel et al., 2025), state tracking (Li et al., 2025; Prakash et al., 2025), and steering model behaviour (Zou et al., 2023; Li et al., 2023; Wu et al., 2024).

**Sparse dictionary learning.** Sparse dictionary learning has emerged as a dominant approach to disentangling the representations of language models into interpretable features, particularly **sparse autoencoders** (SAEs; Bricken et al., 2023; Cunningham et al., 2023; Templeton et al., 2024; Gao et al., 2025a), which learn to reconstruct activations via latent representations in a higher-dimensional sparse space, and **transcoders** (Dunefsky et al., 2024; Ge et al., 2024), which learn to express the nonlinear computation of an MLP in a similar higher-dimensional sparse space. Sparse dictionaries have applications such as steering behaviour (O'Brien et al., 2024; Durmus et al., 2024), data and model diffing (Jiang et al., 2025; Minder et al., 2025), but are often outperformed by supervised methods (Kantamneni et al., 2025; Wu et al., 2025).

**Circuit tracing.** Most approaches to tracing circuits either operate on a coarse granularity or use sparse dictionaries, out of a belief that the neuron basis is uninterpretable. The former category includes early work which uses interchange interventions on coarse components like Wang et al. (2023) and ACDC (Conmy et al., 2023) and other approaches which use gradient-based attribution instead (Syed et al., 2024; Hanna et al., 2024; Mueller et al., 2025). The latter category includes Marks et al. (2025); Ge et al. (2024); Dunefsky et al. (2024); Ameisen et al. (2025); Lindsey et al. (2025).

## 3. Background

We first introduce some necessary definitions for circuit tracing and background on how to evaluate circuits per benchmarks such as Marks et al. (2025); Mueller et al. (2025).

### 3.1. Circuit tracing preliminaries

**Transformer bases.** A Transformer language model $M$ consists of $L$ layers of Transformer blocks, each of which has a sequence-mixing attention block followed by a state-mixing MLP block, with residual connections. The LM takes as input a sequence of $n$ tokens $\mathbf{x} = (x_1, \ldots, x_n)$ and embeds them into input representations $\mathbf{e} = (\mathbf{e}_1, \ldots, \mathbf{e}_n)$, where $\mathbf{e}_j \in \mathbb{R}^{d_{\text{model}}}$. At layer $i \in \{1, \ldots, L\}$, there is a Transformer block that computes the following:

- *Attention output*: $\mathbf{a}^{(i)} = (\mathbf{a}_1^{(i)}, \ldots, \mathbf{a}_n^{(i)})$, where $\mathbf{a}_j^{(i)} \in \mathbb{R}^{d_{\text{model}}}$ is the output of multi-head attention.
- *MLP activations*: $\mathbf{h}^{(i)} = (\mathbf{h}_1^{(i)}, \ldots, \mathbf{h}_n^{(i)})$, where $\mathbf{h}_j^{(i)} \in \mathbb{R}^{d_{\text{ffn}}}$ are the pre-down projection hidden activations within the MLP.
- *MLP output*: $\mathbf{m}^{(i)} = (\mathbf{m}_1^{(i)}, \ldots, \mathbf{m}_n^{(i)})$, where $\mathbf{m}_j^{(i)} \in \mathbb{R}^{d_{\text{model}}}$ is the output of the MLP block.
- *Residual stream*: $\mathbf{r}^{(i)} = (\mathbf{r}_1^{(i)}, \ldots, \mathbf{r}_n^{(i)})$, where $\mathbf{r}_j^{(i)} \in \mathbb{R}^{d_{\text{model}}}$ is a running sum of outputs of all components so far: $\mathbf{r}^{(i)} = \mathbf{r}^{(i-1)} + \mathbf{a}^{(i)} + \mathbf{m}^{(i)}$ with $\mathbf{r}^{(0)} = \mathbf{e}$.

Finally, the model produces output logits $\mathbf{y} = (\mathbf{y}_1, \ldots, \mathbf{y}_n)$ where $\mathbf{y}_j \in \mathbb{R}^{d_{\text{vocab}}}$. For convenience, we refer to the input embeddings $\mathbf{e}$ as $\mathbf{r}^{(0)}$ and the output logits $\mathbf{y}$ as $\mathbf{r}^{(L+1)}$ when discussing circuits over these representations.

**SAE bases.** Sparse autoencoders (SAEs; Bricken et al., 2023; Cunningham et al., 2023) are a dictionary learning technique which decompose Transformer representations into sparse feature bases. Given a representation $\mathbf{x} \in \mathbb{R}^d$, an SAE produces feature activations $\mathbf{f} = g(\mathbf{W}_{\text{enc}}(\mathbf{x} - \mathbf{b}_{\text{pre}}) + \mathbf{b}_{\text{enc}})$ where $\mathbf{f} \in \mathbb{R}^{d_{\text{sae}}}$, $\mathbf{W}_{\text{enc}} \in \mathbb{R}^{d_{\text{sae}} \times d}$, and $g$ is a nonlinearity (traditionally ReLU; Bricken et al., 2023). Many architectural variants exist (Rajamanoharan et al., 2024a; Gao et al., 2025a; Rajamanoharan et al., 2024b). We let $\mathbf{f}^{(i)}$ denote the SAE feature activations for layer $i$.

**Circuits.** A **circuit** is a sparse subgraph $C = (V, E)$ of the computational graph underlying a model's behavior on a specific task or dataset. The **nodes** $V$ are individual computational units (e.g., MLP neurons $h_{j,v}^{(i)}$ or SAE features $f_{j,k}^{(i)}$). We treat the same unit at different token positions $j$ as distinct nodes. The directed weighted **edges** $E$ capture causal influence between nodes in the graph.

Both $V$ and $E$ may be input-dependent: the relevant nodes and their connectivity depend on a given input $\mathbf{x}$.

*Table 1.* Example original inputs with outputs and corresponding counterfactual inputs with outputs from the SVA benchmark.

| Subtask | $x$ | $y$ | $x'$ | $y'$ |
|---|---|---|---|---|
| simple | The **parents** | **are** | The **parent** | **is** |
| within_rc | The athlete that the **managers** | **like** | The athlete that the **manager** | **likes** |
| rc | The **athlete** that the managers like | **does** | The **athletes** that the managers like | **do** |
| nounpp | The **secretaries** near the cars | **have** | The **secretaries** near the cars | **has** |

### 3.2. Evaluating circuits

We use the same procedure for evaluating circuits in all our experiments (including for edge-based ablations). For a circuit $C = (V, E)$, we define $C(x)$ as running the underlying model $M$ with **mean ablation** of the complement of the circuit $\overline{C} = (\overline{V}, \overline{E})$. Mean ablation is an intervention that sets some set of nodes (here, $\overline{V}$) to the mean of their activations over a dataset $\mathcal{D}$, while retaining the remaining computation for nodes not in $\overline{V}$. We denote the activation of a node $v$ on input $x$ as $v(x)$. Formally, we have:

$$C(x) := M(x; \text{do}(v = \mathbb{E}_{d \sim \mathcal{D}}[v(d)]) \text{ for } v \in V) \quad (1)$$

To evaluate a circuit, we follow Marks et al. (2025); Wang et al. (2023) who use two metrics. The first metric, **faithfulness**, says that when the circuit's complement is ablated, the value of the metric $m$ should be close to that of the original model $M$. The second metric, **completeness**, says that ablating the circuit itself should result in a value of the metric $m$ that is close to that of ablating *the entire model* $M$. We normalise by a baseline value $m(\varnothing, \cdot)$ where all nodes in the graph are ablated. We compute the metric over a dataset $\mathcal{D}$ of paired inputs $x$ and outputs $y$. Formally:

$$m(C, x, y) = \ell(y, C(x)) \quad (2)$$

$$\text{Faithfulness}(C) = \frac{\mathbb{E}_{x \sim \mathcal{D}}[m(C, x) - m(\varnothing, x)]}{\mathbb{E}_{x \sim \mathcal{D}}[m(M, x) - m(\varnothing, x)]} \quad (3)$$

$$\text{Completeness}(C) = \frac{\mathbb{E}_{x \sim \mathcal{D}}[m(\overline{C}, x) - m(\varnothing, x)]}{\mathbb{E}_{x \sim \mathcal{D}}[m(M, x) - m(\varnothing, x)]} \quad (4)$$

A perfect circuit thus has faithfulness 1 and completeness 0. Our goal is to identify a sparse circuit (i.e., $|C_V| \ll |M_V|$) that is both faithful and complete.

## 4. Methodology

In our experiments, we seek to compare several possible choices for representing the nodes $V$ in a circuit, including the MLP activations, MLP outputs, attention outputs, and different SAE bases. We evaluate the trade-off between sparsity and faithfulness/completeness on the **subject-verb agreement (SVA)** benchmark in Marks et al. (2025). This standard benchmark provides four simple templatic datasets (simple, rc, within_rc, nounpp), where the goal is to obtain high faithfulness and low completeness when ablating the model's activations. We use and build upon Marks et al.'s codebase for these experiments.

**Datasets.** Each example in the SVA benchmark is a pair of inputs and outputs, where the input is an incomplete sentence with a subject and the output is a verb whose grammatical number matches the number of the subject; see Table 1. The counterfactual input and output is a modification of the original input and output that changes the number of the subject and verb (e.g. from singular to plural). For each task, we take a training set of 300 pairs of original and counterfactual inputs and evaluate on 40 held-out pairs, following Marks et al. (2025).

**Metric.** To measure faithfulness and completeness, we must define the metric $m$. Recall that SVA consists of pairs of original and counterfactual inputs $x, x'$ and outputs $y, y'$ as described in Table 1, and that the model $M(x)$ outputs a vector of next-token logits given input $x$. We take the metric $m$ to be the logit difference:

$$m(C, x) = [C(x)]_y - [C(x)]_{y'} \quad (5)$$

In SVA, $y$ and $y'$ are singular and plural forms of the same verb (e.g. *is* and *are*, as in the example in Table 1).

**Attribution method for selecting $V$.** On each task, we obtain a circuit using the training subset, then evaluate its faithfulness and completeness on a validation subset. We obtain circuits by greedily taking the $k$ highest-attribution nodes. The nodes are individual features, e.g. MLP neurons.

We use Integrated Gradients (IG; Sundararajan et al., 2017) to compute attribution scores (we will introduce another method later). We specifically use IG-activations, proposed in Marks et al. (2025).[1] For a node $v \in V$ with scalar activation value $v(x)$, IG-activations interpolates between the counterfactual input $x'$ and the original input $x$:

$$\text{Attr}_{\text{IGAct}}(v) = \mathbb{E}_{(x, x') \sim \mathcal{D}}[\text{IGact}_v(x; x')] \quad (6)$$

$$\text{IGact}_v(x; x') = \Delta v \int_{\alpha=0}^{1} \frac{\partial m(M, x; \text{do } v = v_\alpha)}{\partial v} \text{d}\alpha \quad (7)$$

$$\approx \Delta v \cdot \frac{1}{n} \sum_{i=1}^{n} \frac{\partial m(M, x; \text{do } v = v^{(i)})}{\partial v} \quad (8)$$

$$\text{where } \Delta v = v(x) - v(x'), \quad v_\alpha = v(x') + \alpha \Delta v,$$

$$v^{(i)} = v(x') + \frac{i}{n} \Delta v \quad (9)$$

---

[1] See §A for discussion on different versions of IG.

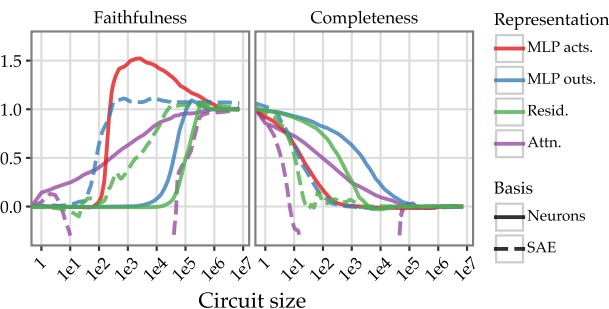

*Figure 1.* Faithfulness and completeness for different choices of representation in the model (residual stream, attention, MLP activations, or MLP outputs) and basis (neurons or SAE) when applying Integrated Gradients, averaged over the 4 SVA tasks with paired data.

**Evaluation and setup.** For each task, we evaluate circuits of varying size $k$ for each choice of feature basis. We evaluate the faithfulness and completeness of the circuits, averaged over the evaluation set. We plot these values against $k$ and identify methods that achieve the highest faithfulness and lowest completeness at the smallest circuit size.

We use the Llama 3.1 8B base model.[2] We compare the neuron basis on MLP outputs, MLP activations, attention outputs, and the residual stream, along with the SAE basis on MLP outputs, attention outputs, and the residual stream. For the SAE basis, we use the 8x width SAEs from Llama Scope (He et al., 2024). In all experiments, we vary the size of the circuit by varying the attribution score threshold. For SAEs, we allow the error term to be included as a node as well if its attribution score exceeds the selected threshold, following the default evaluation setup in Marks et al. (2025).

## 5. MLP activations are a sparse basis for circuits

We now run a series of experiments which show the advantages of using MLP activations over alternatives in the circuit tracing setting.

### 5.1. MLP activations are much sparser than MLP outputs

In Figure 1, we plot circuit size vs. faithfulness and completeness for each of the methods and baselines, averaged over the four tasks. We find that MLP activations yield significantly smaller circuits than MLP outputs (by a factor of $100\times$). Using MLP activations instead of MLP outputs also significantly closes the gap with SAEs.

---

[2]In §E, we report addition results for Gemma 2 2B and Gemma 2 9B with the Gemma Scope SAEs (Lieberum et al., 2024), which add additional evidence to our conclusions.

*Table 2.* Linearised treatments of nonlinear operations.

| Operation | Definition | Linearised Treatment |
|---|---|---|
| RMSNorm | $x_i/\sqrt{\epsilon + \overline{x^2}}$ | $x_i/\text{Freeze}(\sqrt{\epsilon + \overline{x^2}})$ |
| SiLU | $x_i \cdot \sigma(x_i)$ | $x_i \cdot \text{Freeze}(\sigma(x_i))$ |
| Attention | $\sum_k A_{qk} v_k$ | $\sum_k \text{Freeze}(A_{qk}) v_k$ |

Conceptually, we hypothesize that MLP activations work better because the activation coordinates are a privileged basis (due to the element-wise nonlinearity), whereas the MLP outputs are not.[3] We present further analysis in §F.

### 5.2. RelP closes the gap between MLP activations and MLP SAEs

When using IG to compute attribution scores, we still find neurons to be somewhat less sparse than SAE features. Additionally, IG can be inefficient to compute due to the need for multiple backward passes, and imprecise due to the need to numerically estimate it with samples (Makino et al., 2024). We therefore use an alternative methods which only requires a single backward pass: RelP (Jafari et al., 2025). Building on prior work, we are the first to apply RelP directly to the individual MLP neurons and to compute neuron-to-neuron edge weights.[4]

To score the causal importance of a component with RelP, we apply gradient-based attribution to a **replacement model**, wherein all nonlinearities are replaced with linear approximations such that the model remains locally faithful to the original on a specific input (Ali et al., 2022; Ge et al., 2024; Ameisen et al., 2025). While this modifies the backward pass computation, it retains the same forward pass as the original model. Importantly, **our final evaluations are performed on the original model**.

We use the local replacement rules listed in Table 2 for nonlinearities in the Llama 3 architecture. Additionally, we use the **half rule** from layerwise relevance propagation (LRP), which divides the gradient by two through multiplicative interactions (e.g. the elementwise multiplication in gated MLPs). This ensures the *completeness* property, i.e. that the total attribution score is conserved layer-by-layer (Arras et al., 2019; Achtibat et al., 2024; Jafari et al., 2024, and see proof in §B)

To compute the attribution score for a node $v \in V$ when processing input $x$, we multiply the activation $v(x)$ by the

---

[3]Per Elhage et al. (2021), a *privileged basis* is one where the architecture encourages features to align with the basis dimensions by applying non-rotation-invariant transforms.

[4]We concurrently developed the RelP attribution method along with Jafari et al. (2025) did, but we keep the name RelP throughout this work for simplicity.

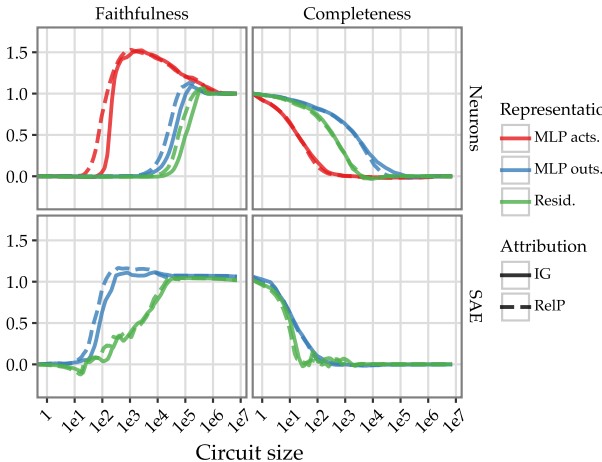

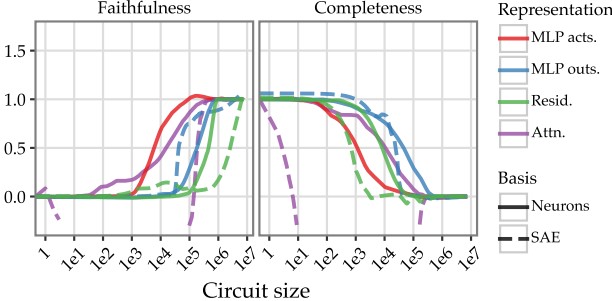

*Figure 3.* Faithfulness and completeness for different choices of representation and basis when applying Integrated Gradients, averaged over the 4 SVA tasks with **unpaired** data.

*Figure 2.* Faithfulness and completeness for Integrated Gradients vs. RelP, for different choices of representation in the model and basis (neurons or SAE), averaged over the 4 SVA tasks with paired data

gradient of the metric $m$ (based on the replacement model $M_{\text{replacement}}$) with respect to $v$:

$$\text{Attr}_{\text{RelP}}(v) = \mathbb{E}_{(x,x') \sim \mathcal{D}}[\text{RelP}_v(x; x')] \quad (10)$$

$$\text{RelP}_v(x; x') = \Delta v \frac{\partial\, m(M_{\text{replacement}}, x)}{\partial\, v(x)} \quad (11)$$

We implement all of our modifications by overwriting the backward pass of the relevant components to detach the nonlinearities and call `torch.autograd.grad` on saved activations to compute gradients.

We run the same experiments as §5.1 on the SVA dataset, but this time compare results given by RelP with those by IG. We plot results in Figure 2.

RelP outperforms IG in almost all settings. For MLP activations, performance reaches near-perfect faithfulness and completeness with only $\sim 200$ neurons. RelP also improves faithfulness on MLP outputs and the residual stream. Finally, RelP slightly improves performance for SAEs trained on MLP outputs, but not for residual stream SAEs. Note that IG is computed with 10 backward passes in our setup, while RelP requires only one backward pass. Additionally, in §C, we confirm that RelP is the best-performing attribution method on the Mechanistic Interpretability Benchmark. RelP therefore **closes the gap between MLP neuron sparsity and SAE sparsity**.

### 5.3. The same findings hold on unpaired data

Our results so far rely on templatic pairs to compute attribution scores on the training set. Real-world data is non-templatic and messy; for many interesting behaviours, we may not have hypotheses that allow us to generate paired data. Tracing circuits in this more realistic setting is impor-

tant for scaling up interpretability.

We thus consider an *unpaired* circuit-finding task. Instead of using a counterfactual input $x'$ as the baseline to compute IG and RelP for each circuit component, we use a zero baseline. Formally, this gives us:

$$\text{IGAct}_v(x) = v(x) \int_{\alpha=0}^{1} \frac{\partial m(M, x; \text{do } v = \alpha v(x)))}{\partial v(x)} \mathrm{d}\alpha$$

$$\approx v(x) \frac{1}{n} \sum_{i=1}^{n} \frac{\partial m(M, x; \text{do } v = \frac{i}{n} v(x)))}{\partial v(x)}$$

$$\quad (12)$$

$$\text{RelP}_v(x) = v(x) \frac{\partial\, m(M_{\text{replacement}}, x)}{\partial\, v(x)} \quad (13)$$

We keep evaluation identical to the paired setting (i.e. we still compute the difference between original and counterfactual logits as the metric). We test **mean ablation** during evaluation (as in Equation (1)). We report results with **zero ablation** (setting ablated nodes to have activation of 0, which is simpler but may result in out-of-distribution activations; Li & Janson, 2024) in §D.

We run experiments on the SVA dataset, but provide only the original (not counterfactual) input for training. The results are plotted in Figure 3: we again observe that the MLP activation neuron basis requires a considerably smaller circuit to achieve good faithfulness and completeness relative to other methods.

Finally, we evaluate our approach on the unpaired setting with mean ablation in Figure 4. We observe improvements in both faithfulness and completeness over IG, with RelP requiring fewer neurons to achieve good performance.

### 5.4. RelP finds more faithful edges than IG

Our results so far have shown that the MLP activations are a sparse basis for circuits, and that RelP can find a better set of neurons than IG. However, a circuit is not just a set of neurons, but also **the edges connecting them**. We now

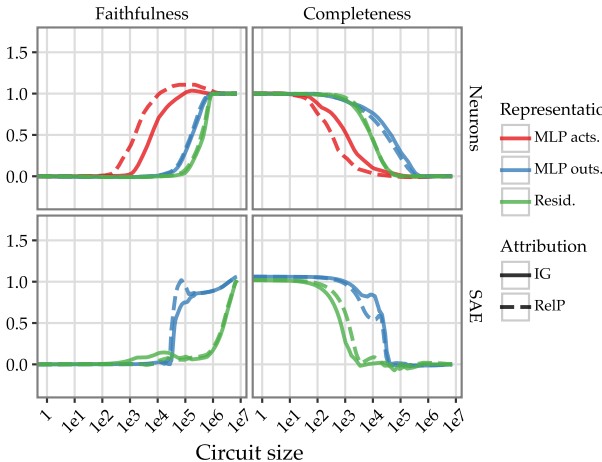

*Figure 4.* Faithfulness and completeness for Integrated Gradients vs. RelP, for different choices of representation in the model and basis (neurons or SAE), averaged over the 4 SVA tasks with **unpaired** data

evaluate various methods for computing edge weights.

**Preliminaries.** We can use either IG or our method to compute edge weights. Given a source node $v_s$ and a target node $v_t$, the attribution score for each method is formally expressed (in the unpaired setting) as:

$$\text{IGAct}_{v_s \to v_t}(x) = v_s(x) \int_{\alpha=0}^{1} \frac{\partial v_t(x; \text{do } v_s = \alpha v_s(x))}{\partial v_s(x)} d\alpha \tag{14}$$

$$\approx v_s(x) \frac{1}{n} \sum_{i=1}^{n} \frac{\partial v_t(x; \text{do } v_s = \frac{i}{n} v_s(x))}{\partial v_s(x)} \tag{15}$$

$$\text{RelP}_{v_s \to v_t}(x) = v_s(x) \frac{\partial v_t^{\text{replacement}}(x)}{\partial v_s^{\text{replacement}}(x)} \tag{16}$$

In order to make the edge weight interpretable in the context of the attribution graph, we normalise it by the total attribution score of the target neuron, which we term the **attribution flow** via this edge:

$$\text{Flow}_{v_s \to v_t}^{\text{IGAct}} = \frac{\text{IGAct}_{v_s \to v_t}(x)}{v_t(x)} \text{IGAct}_{v_t}(x) \tag{17}$$

$$\text{Flow}_{v_s \to v_t}^{\text{RelP}} = \frac{\text{RelP}_{v_s \to v_t}(x)}{v_t(x)} \text{RelP}_{v_t}(x) \tag{18}$$

This tells us how much of the final logits can be attributed to the path(s) from the source neuron to the target neuron. A useful property of this value is that it normalises for the sign of $v_t(x)$, which need not correspond to the sign of $\text{IGAct}_{v_t}(x)$ or $\text{RelP}_{v_t}(x)$.

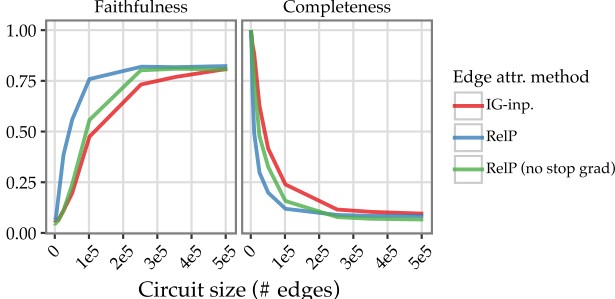

*Figure 5.* Faithfulness and completeness for edge-based circuit evaluation on the SVA benchmark. All methods use MLP activations as the neuron basis. Circuits are pruned by removing edges based on attribution scores, with neurons removed when all incoming or outgoing edges are pruned.

**Methodology.** We compute edge attribution using MLP activations as our feature basis. We start with the top $10^3$ neurons to keep edge evaluation tractable, yielding up to $5 \cdot 10^5$ potential edges per example. For each pair of neurons in our filtered set, we compute edge weights using the attribution flows defined in Equation (18).

We compare three edge attribution methods:

- **IG-inp.**: The integrated gradients variant described above (see Hanna et al., 2024), using 10 steps.
- **RelP**: Our gradient-based method with stop-gradients and straight-through estimation, including additional stop-gradients applied to intermediate MLP layers.
- **RelP (no stop grad on MLPs)**: Same as above, but without the additional stop-gradients on intermediate MLP layers.

RelP without stop-grads on intermediate MLPs gives the *total effect* of the source neuron on the target neuron via paths through other intermediate neurons. Applying stop-grads on intermediate MLPs computes the *direct effect* instead.

We keep the top $k$ edges by magnitude, varying $k$ to select some percent of edges. A neuron is removed from the circuit if none of its incoming edges or none of its outgoing edges are excluded. We use the same evaluation as in §4.

**Result.** The results in Figure 5 show that **RelP with stop grads** achieves the best performance of all methods, reaching over 80% faithfulness while maintaining high completeness with only $\approx 10^5$ edges (10% of the candidate edges). RelP Pareto-dominates both alternatives.

## 6. Case study: Multi-hop reasoning

We now use neuron-level circuit tracing to investigate how Llama 3.1 8B Instruct performs a simple multi-hop reasoning task for retrieving state capitals, following Lindsey et al. (2025). We trace circuits for each example in the dataset

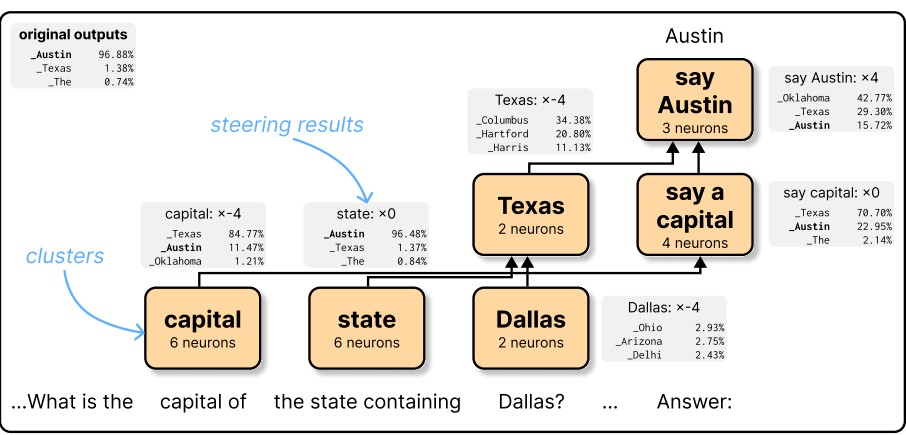

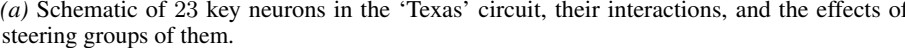

| Cluster | Neurons |
|---|---|
| Capital | L3/N14335− (English-specific), L4/N13489− (multilingual), L19/N2520− (Washington, D.C.), L20/N3520+, L16/N13326−, L13/N4038+ |
| State | L0/N9296− (English-specific), L2/N5246+ (multilingual), L4/N604− (broader semantics), L19/N4478+ (statehood), L21/N5790−, L21/N12118− |
| Dallas | L0/N12136− (primarily Houston), L5/N8659+ (various Texas locations) |
| Texas | L6/N10965−, L21/N3093+ |
| Say a capital | L23/N8079−, L21/N4924−, L23/N2709−, L17/N3663+ (all specifically include "capital" in their description) |
| Say Austin | L30/N8371+ (words ending in "un"), L31/N4876+, L31/N6705+ |

*(a)* Schematic of 23 key neurons in the 'Texas' circuit, their interactions, and the effects of steering groups of them.

*(b)* Key neurons in the 'Texas' circuit, clustered into 6 groups.

*Figure 6.* The 'Texas' circuit.

and analyse the resulting graphs.

We examine three additional tasks in §H, which we do not present in the main text since they only add additional evidence to the same claims we make here. Lindsey et al. (2025) use cross-layer transcoders (CLTs) to trace circuits for three of those tasks. We aim to show that we can find comparably interpretable circuits in the MLP neuron basis.

**Task.** Lindsey et al. (2025) study a state capitals task with questions such as "What is the capital of the state containing Dallas?" The goal is to isolate circuit components that are responsible for each reasoning hop.

We construct a simple dataset of 50 multi-hop reasoning questions involving state capitals, which uses the same question style as Lindsey et al., but is reformatted for the chat-tuned model we study. An example is shown below:

> **Multi-hop reasoning on state capitals**
>
> **User**  What is the capital of the state containing Dallas?
>
> **Assistant**  Answer: _Austin

**Circuit tracing methodology.** We follow Ameisen et al. (2025) and use the total value of the top-$k$ next-token logits as the metric to attribute from, where $k = 5$ unless otherwise stated: $m(M, x) = \sum_{i=1}^{k} [M(x)]_i$. After computing **unpaired** RelP for each node $v$ on an example $x \in \mathcal{D}$, we filter for nodes that meet some attribution threshold $\tau$ relative to the total logit value:

$$V(x) = \{v \in V : \mathrm{RelP}_v(x) \geq \tau \cdot m(M, x)\} \quad (19)$$

We set $\tau$ to be 0.005 in our experiments. We use RelP to attribute nodes and RelP with stop-grads to attribute edges. We also manually exclude 12 neurons from our circuits; see §G.1.

**Neuron descriptions.** For the MLP neurons in Llama 3.1 8B Instruct, we have automatic neuron descriptions from Choi et al. (2024), which we can use to interpret our neuron circuits. In that work, the best of 20 LM-generated descriptions were chosen based on how well a finetuned simulator could predict the ground-truth activations of that neuron given a proposed natural-language description.

**Steering.** Given a set of neurons $V$, we can fix the activations of these neurons to be a scalar multiple $\alpha$ of their original activations on input $x$, and measure the change in the model's output. Formally:

$$M^{\mathrm{Steer}(V, \alpha)}(x) = M(x; \mathrm{do}\, v = \alpha v(x) \text{ for } v \in V) \quad (20)$$

Steering allows us to understand how the neurons in $V$ causally contribute to the model's behaviour on this input.

### 6.1. Examining the 'Texas' circuit

In the example input shown before, the model must perform the multi-hop reasoning chain "Dallas → Texas → Austin". We performed automatic circuit tracing on the input to recover a subset of 257 total neurons with high attribution scores, then manually identified a subset of 23 neurons with particularly meaningful descriptions; the circuit is schematically depicted in Figure 6. These neurons, listed in Figure 6b, cluster into six groups that match the categories from Lindsey et al. (2025). This is striking, since we are investigating a different model (Llama 3.1 8B Instruct) and a different set of representations (the neuron basis).

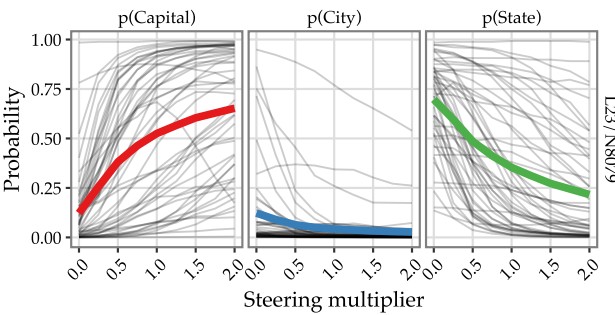

*Figure 7.* Effect of steering `L23/N8079-` on the capitals dataset.

We steer each cluster of neurons to check whether the resulting effect on the model's outputs corresponds with the hypothesised role of each cluster (e.g. steering the "say a capital" cluster negatively should cause the model to not output capital cities). We show the top next-token predictions after steering each cluster for the Texas example (selecting a single $\alpha \in [-4, 4]$) in the figure.

All clusters except for "state" change the model's top prediction when steered. The changes to model output correspond to the hypothesised role of each cluster; for example, suppressing the "Texas" cluster causes the model to still output state capitals but for other states.

### 6.2. Steering a single "say a capital" neuron

We now turn to the whole dataset and examine the highest-attribution neuron in the "say a capital" cluster: `L23/N8079-`, with the description *the phrase "is" when referring to state capitals*. We investigate whether this neuron plays a consistent causal role across the dataset by steering it with $\alpha \in \{0, 0.25, \dots, 2\}$. For each example, we measure the resulting output probability for the **capital** (e.g. "Austin"; the correct answer), the **state** ("Texas"), and the original non-capital **city** ("Dallas"). We plot the resulting probabilities for each type of answer against $\alpha$ in Figure 7.

This single neuron can be steered to flip the top output from the capital to the state in a majority of examples. This validates that individual neurons can play a significant and interpretable role in model behaviour; this neuron is evidence for the reasoning step of mapping states to their capitals being implemented in hidden states rather than e and that our neuron-level circuit tracing algorithm is able to discover them without sparse dictionaries.

## 7. Discussion

Our findings have several implications for the future of circuit tracing, and interpretability in general.

**Neurons as a practical avenue for interpretability.** The neuron basis has not been sufficiently explored in the circuits literature, meaning the amount of uplift from SAEs is unknown. For example, Ameisen et al. (2025) do not report the minimal ablation of their gradient-based tracing algorithm where CLTs are replaced with neurons; instead, they only report an ablation where neurons are thresholded by activation. Despite earlier use of the MLP activations in interpretability research (Bricken et al., 2023; Gurnee et al., 2023), recent circuits work has not used them as a baseline. **New sparse dictionary-learning methods ought to include an MLP neuron-level comparison**.

SAEs plausibly make unjustified assumptions about the geometry and frequency of features (Hindupur et al., 2025; Chanin et al., 2025) with unclear benefits from doing so (Wu et al., 2025; Kantamneni et al., 2025). Until we better understand what properties the true feature basis has, it seems prudent to exhaust the neuron basis first. Alternative techniques without reconstruction error (Shafran et al., 2025) are also compatible with our approach.

**Architectural trends favour MLP sparsity.** One explanation for why the MLP activations are sparse is that sparsity is a desirable property in Transformer MLPs which is being selected for by architectural changes. The switch to gated MLPs may have enabled packing more features without sacrificing sparsity. Furthermore, mixture-of-experts MLPs explicitly enforce sparsity by routing inputs to a subset of expert MLPs. Architectural sparsity is known to be valuable for interpretability (Gao et al., 2025b). If existing architectures are tending towards greater sparsity, interpretability research can take advantage of this.

**Limitations.** Firstly, there are still too many neurons in a reasonably comprehensive circuit for a human being to easily interpret. We need principled approaches for **clustering neurons** in a circuit, as well as improving **automatic natural-language descriptions** of neurons and neuron clusters. Secondly, our implementation could be made more efficient, since it relies on serial calls to the `torch.autograd.grad` function which leads to low compute utilisation. However, we do not need to load SAEs into memory, so our approach is tractable for large models. We address these limitations to a great extent in Arora et al. (2026), with an end-to-end automated circuit tracing pipeline.

## 8. Conclusion

We have shown that neuron-level circuit tracing on the MLP activations can achieve the same performance as SAE circuits with equivalent sparsity. Our case studies further demonstrate that our neuron-basis circuits can be used to

reproduce findings from prior work using CLTs (Ameisen et al., 2025; Lindsey et al., 2025) as well as new results on user modelling. We also show that these results are complementary to automated natural-language descriptions of neurons. We hope that this work renews attention to the MLP activations as a potentially interpretable basis for circuit tracing.

## Acknowledgements

We thank Christopher Potts, Dan Jurafsky, Samuel Marks, Achyuta Rajaram, Harshit Joshi, and Aaron Mueller for feedback on an earlier version of this draft. We thank Christopher D. Manning, Dami Choi, Vincent Huang, Neil Chowdhury, and researchers from the Stanford NLP group and the Stanford #weekly-interp-meeting for helpful discussion throughout the project.

## Impact Statement

The goal of this work is to advance the interpretability of language models. By introducing an efficient and scalable technique for tracing the internal computations of a model, we democratise AI safety research which is useful for independent auditors and may allow mitigating harmful behaviours by these models. However, this tool may also enable intervening on model internals to produce harmful outputs. On balance, we believe this research is societally beneficial since it increases the transparency of powerful AI systems.

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

# Appendix

# A. Different operationalisations of IG on internals

Integrated Gradients (Sundararajan et al., 2017) was originally defined as a technique to attribute outputs to inputs in a deep neural network, via approximating the following path integral with a Reimann sum:

$$\mathsf{IG}_i(x) = (x_i - x_i') \int_{\alpha=0}^{1} \frac{\partial F(x' + \alpha(x - x'))}{\partial x_i} \mathrm{d}\alpha \tag{21}$$

$$\approx (x_i - x_i') \frac{1}{n} \sum_{i=1}^{n} \frac{\partial F(x^{(i)})}{\partial x_i} \tag{22}$$

$$\text{where } x^{(i)} = x' + \frac{i}{n}(x - x') \tag{23}$$

where attributions to each index of the input $i$ are computed in parallel, and the interpolation with $\alpha$ is done simultaneously for all inputs. We define $x^{(i)} = x' + \frac{i}{n}(x - x')$, i.e. discrete interpolation between $x$ and $x'$.

We leave justification of this technique to prior literature; briefly, it arises as one solution to a set of axioms which encode useful properties for attribution methods to have.

However, the lack of application of IG to internals in the original paper has led to a profusion of differing techniques for internals attribution, which need not satisfy the original IG axioms. We introduce each of these below and benchmark them in later sections.

**Conductance.** Proposed in a follow-up paper to IG (Dhamdhere et al., 2019), conductance is arguably the most principled application of IG to internals; it simply uses the chain rule to compute the portion of the gradients that flow via some internal component $y$ while not changing any other aspect of the IG path integral:

$$\mathsf{Cond}_i^y(x) = (x_i - x_i') \int_{\alpha=0}^{1} \frac{\partial F(x' + \alpha(x - x'))}{\partial y} \frac{\partial y}{\partial x_i} \mathrm{d}\alpha \tag{24}$$

where the total conductance is a sum of this value over all $x_i$. While this seems expensive to compute at first glance (you must compute the Jacobian from outputs to $y$ and from $y$ to $x$, which will use a lot of memory if you want to parallelise), a later work (Shrikumar et al., 2018) algebraically simplified the above expression and found a much cheaper but equivalent expression:

$$\mathsf{Cond}_y(x) = \sum_i \mathsf{Cond}_i^y(x) \tag{25}$$

$$= \int_{\alpha=0}^{1} \frac{\partial F(x' + \alpha(x - x'))}{\partial(y' + \alpha(y - y'))} \frac{\partial(y' + \alpha(y - y'))}{\partial\alpha} \mathrm{d}\alpha \tag{26}$$

$$\approx \sum_{i=1}^{n} \frac{\partial F(x^{(i)})}{\partial y} (F_y(x^{(i)}) - F_y(x^{(i-1)})) \tag{27}$$

Unfortunately, conductance is not used in recent benchmarks of gradient-based attribution; subtly different techniques have been proposed and adopted recently, which we turn to now.

**IG-inputs.** Hanna et al. (2024) introduce a nearly identical technique to conductance but seemingly arrived at it independently. The only difference is the path integral is misspecified; rather than taking into account the possibly variable step size of $y$ when interpolating $x$, they average the gradient of $y$ *first* and multiply outside by the total step size over the entire path. If step size of $y$ is relatively constant then this is a reasonable empirical approximation of conductance.

$$\mathsf{IGinp}_y(x) \approx (F_y(x) - F_y(x')) \sum_{i=1}^{n} \frac{\partial F(x^{(i)})}{\partial y} \tag{28}$$

**IG-activations.** Marks et al. (2025) introduce a version of IG where the interpolation is done over $y$ rather than the inputs $x$, but otherwise the attribution scoring is unchanged . This is essentially IG but pretending that $y$ is an input variable.

$$\mathsf{IGact}_y(x) \approx (F_y(x) - F_y(x')) \sum_{i=1}^{n} \frac{\partial F(x; \mathrm{do}\, y = y^{(i)})}{\partial y} \tag{29}$$

# B. Justification for the half rule in RelP

Integrated Gradients (Sundararajan et al., 2017) was designed to satisfy a few axioms which ensure the fairness of the attribution scores. Two of these axioms which are particularly important are **completeness** (i.e. the sum of the attribution scores of the inputs to a function equals the function's output) and **sensitivity** (if only a single input is changed and this affects the output, the attribution scores will change). Completeness implies sensitivity. Here, we show that RelP also satisfies completeness (as long as the half rule is applied to the gated MLP) without the need for numerical integration with multiple backward passes.

To begin with, we describe how the replacement model is constructed. When we modify the model following the RelP procedure, we apply the following changes to the model's components:

**Linearised components.** For multi-head attention, we similarly detach the entire QK-path $\mathrm{softmax}(\mathrm{QK}^\top/\sqrt{d_k})$ and treat it as a constant elementwise multiplier $\mathbf{A}$ on the value activations. (Here, $j$ indicates the query token position.)

$$\mathrm{MHA}_j^{(i)}(\mathbf{x}) = \mathrm{W}_{\mathrm{out}}^{(i)}\left(\mathrm{Concat}_h\left(\mathbf{A}_j^{(i,h)} \cdot \mathrm{W}_{\mathrm{value}}^{(i,h)}\mathbf{x}\right)\right) \tag{30}$$

For the RMSNorms when reading from the residual stream, as well as the final layer before the unembedding, we again detach the normalisation and treat it as a dimension-wise constant multiplier $\mathbf{r}$.

$$\mathrm{RMSNorm}^{(i,j)}(\mathbf{x}) = \mathbf{r}^{(i,j)} \odot \mathbf{x} \tag{31}$$

It is clear that for these two, the function is completely linear, and thus $\partial f(\mathbf{x})/\partial \mathbf{x}$ is a constant, and so input times gradient retains the completeness property of IG. In fact, IG and RelP are equal on linear functions, since their gradient is constant:

$$\mathrm{IG}_{\mathbf{x}}(f) = \mathbf{x}\int_{\alpha=0}^1 \frac{\partial f(\alpha\mathbf{x})}{\partial \mathbf{x}}d\alpha \tag{32}$$

$$= \mathbf{x}\frac{\partial f(\mathbf{x})}{\partial \mathbf{x}} \tag{33}$$

**Bilinearised MLP.** For the gated MLP (as used in most modern Transformer language models), we detach the SiLU nonlinearity's effect and treat it as a constant elementwise multiplier $\mathbf{s}$ on the gate activation.

$$\mathrm{MLP}_j^{(i)}(\mathbf{x}) = \mathrm{W}_{\mathrm{down}}^{(i)}\left(\mathbf{s} \odot \mathrm{W}_{\mathrm{gate}}^{(i)}\mathbf{x} \odot \mathrm{W}_{\mathrm{up}}^{(i)}\mathbf{x}\right) \tag{34}$$

Here, the elementwise multiplication means that the MLP is actually *bilinear* after our modification, so the completeness property is not satisfied by naïve input times gradient.[5] Consider the gradient of the MLP activations $h(\mathbf{x}) = \mathbf{s} \odot \mathrm{W}_{\mathrm{gate}}^{(i)}\mathbf{x} \odot \mathrm{W}_{\mathrm{up}}^{(i)}\mathbf{x}$ with respect to the input $\mathbf{x}$:

$$\frac{\partial h}{\partial \mathbf{x}} = \mathrm{diag}(\mathbf{s})[\mathrm{diag}(\mathrm{W}_{\mathrm{up}}\mathbf{x})\mathrm{W}_{\mathrm{gate}} + \mathrm{diag}(\mathrm{W}_{\mathrm{gate}}\mathbf{x})\mathrm{W}_{\mathrm{up}}] \tag{35}$$

If we apply naïve input times gradient to this expression, we get:

$$\frac{\partial h}{\partial \mathbf{x}}\mathbf{x} = \mathrm{diag}(\mathbf{s})[\mathrm{diag}(\mathrm{W}_{\mathrm{up}}\mathbf{x})\mathrm{W}_{\mathrm{gate}}\mathbf{x} + \mathrm{diag}(\mathrm{W}_{\mathrm{gate}}\mathbf{x})\mathrm{W}_{\mathrm{up}}\mathbf{x}] \tag{36}$$

$$= \mathrm{diag}(\mathbf{s})[\mathrm{W}_{\mathrm{up}}\mathbf{x} \odot \mathrm{W}_{\mathrm{gate}}\mathbf{x} + \mathrm{W}_{\mathrm{gate}}\mathbf{x} \odot \mathrm{W}_{\mathrm{up}}\mathbf{x}] \tag{37}$$

$$= \mathbf{s} \odot 2(\mathrm{W}_{\mathrm{up}}\mathbf{x} \odot \mathrm{W}_{\mathrm{gate}}\mathbf{x}) \tag{38}$$

---

[5]Pearce et al. (2025) has related discussion on the interpretability benefits from using bilinear MLPs as replacements for conventional MLPs with nonlinear activation functions.

$$= 2h(\mathbf{x}) \tag{39}$$

Another way to derive this is to use the fact that the replacement MLP is a homogeneous function of degree 2 (i.e. $\mathrm{MLP}(\alpha \mathbf{x}) = \alpha^2 \mathrm{MLP}(\mathbf{x})$, under RelP linearisation). Euler's theorem[6] asserts that for a homogenous function of degree $k$, the following partial differential function holds:

$$k f(\mathbf{x}) = \sum_{i=1}^{n} \mathbf{x}_i \frac{\partial f(\mathbf{x})}{\partial \mathbf{x}_i}(\mathbf{x}) \tag{40}$$

Both these results directly motivate the half rule for the gated MLP, wherein we divide the gradient by 2 through multiplicative interactions in order to preserve completeness; otherwise, any time a path passes through a gated MLP, its attribution will be doubled.

Therefore, with the half rule on the gated MLP elementwise multiplication, we have confirmed that RelP satisfies completeness (and thus sensitivity) without the need for numerical integration.

---

[6]https://en.wikipedia.org/wiki/Homogeneous_function#Euler's_theorem

## C. Comparing attribution methods on the MIB benchmark

The Mechanistic Interpretability Benchmark (MIB) (Mueller et al., 2025) includes a **circuit localisation** track which tests the ability of attribution methods to find a subnetwork of the model responsible for a specific behaviour. MIB provides IG baselines adapted to both node- and edge-based attribution, allowing us to validate RelP on both node and edge importance scoring since it is a drop-in replacement for IG. However, unlike the feature-level evals so far, MIB finds a graph over *much larger* components of the model (attention heads and MLPs), so these results are on a less granular setting than the SVA results in §5.

**Methodology.** We adapt MIB's IG-based baselines to use `RelP` by applying our stop-gradients and straight-through handling directly to the model. All methods use counterfactual (CF) ablations: non-included nodes or edges are ablated by substituting their activations with those from a counterfactual example, identical to SVA. We replace the gradient computation in MIB's two best attribution methods: `EAP (CF)` (Nanda, 2023; Syed et al., 2024), which linearly approximates the indirect effect (IE; Pearl, 2001) for all nodes or edges, and `EAP-IG-inp. (CF)` (Hanna et al., 2024), which improves on `EAP (CF)` by performing multiple steps, trading speed for approximation quality. We use identical hyperparameters to the original baselines, only substituting `RelP` for gradient computation.[7] We report results for the best-performing RelP variant.

**Metrics.** MIB provides two metrics, building on faithfulness as defined in Equation (3):

$$\text{CPR} = \int_{k=0}^{1} \text{Faithfulness}(C_k)\mathrm{d}k \tag{41}$$

$$\text{CMD} = \int_{k=0}^{1} |1 - \text{Faithfulness}(C_k)|\mathrm{d}k \tag{42}$$

where $C_k$ is the circuit containing proportion $k$ of the model's edges. In practice, these integrals are approximated using Riemann sums over discrete values of $k$. We report CMD scores, where lower values indicate better performance. Intuitively, CMD measures how closely a circuit's behavior resembles the full model's task-specific behavior across different circuit sizes.

**Results.** We evaluate `RelP` on MIB for Llama 3.1, averaging results across three runs with different random seeds. Results for existing methods are taken from the original MIB paper.

*Table 3.* MIB circuit localization results for Llama 3.1 models, CMD scores. Lower scores indicate better performance. RelP (highlighted) represents our method using the replacement model with straight-through gradients.

| Method | IOI | Arithmetic | MCQA | ARC (E) | ARC (C) | Avg |
|---|---|---|---|---|---|---|
| Random | 0.74 | 0.75 | 0.74 | 0.74 | 0.74 | 0.74 |
| EAP (mean) | 0.04 | 0.07 | 0.16 | 0.28 | 0.20 | 0.15 |
| EAP (CF) | **0.01** | 0.01 | **0.09** | **0.11** | 0.18 | **0.08** |
| EAP-IG-inp. (CF) | **0.01** | **0.00** | 0.14 | **0.11** | 0.22 | 0.10 |
| EAP-IG-act. (CF) | **0.01** | **0.00** | 0.13 | 0.30 | 0.37 | 0.16 |
| NAP (CF) | 0.29 | 0.28 | 0.32 | 0.69 | 0.69 | 0.45 |
| NAP-IG (CF) | 0.19 | 0.18 | 0.33 | 0.67 | 0.67 | 0.41 |
| IFR | 0.83 | 0.22 | 0.48 | 0.64 | 0.76 | 0.59 |
| RelP (ours) | **0.01** | **0.00** | 0.11 | 0.15 | **0.15** | **0.08** |

In Table 3, we find that `RelP` achieves the best CMD score on three tasks, and is the second-best on the remaining two tasks. `RelP` thus works well for edge-based attribution between larger modules, in addition to fine-grained features.

---

[7]Note that `EAP (CF)` differs from `EAP-IG-inp. (CF)` with step size of 1: `EAP (CF)` computes gradients at the original input, whereas `EAP-IG-inp. (CF)` with steps=1 computes gradients at the counterfactual input due to its interpolation scheme.

# D. Unpaired SVA results with zero ablation

The overall results hold in the unpaired setting when using zero ablation: MLP activations are the sparsest representation and RelP outperforms IG.

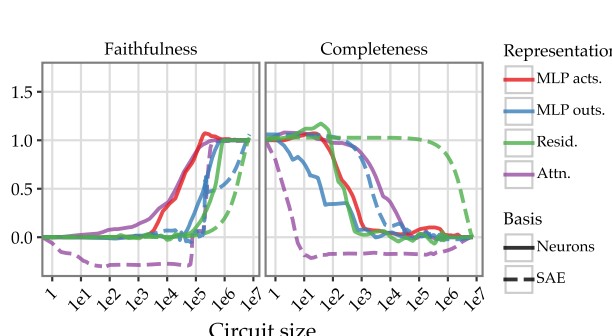

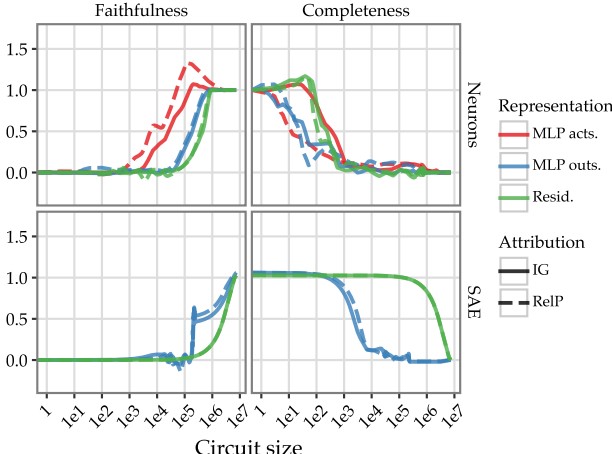

*(a)* Faithfulness and completeness for different choices of representation and basis when applying Integrated Gradients, averaged over the 4 SVA tasks with **unpaired data** under **zero ablation**.

*(b)* Faithfulness and completeness for Integrated Gradients vs. RelP, for different choices of representation in the model and basis (neurons or SAE), averaged over the 4 SVA tasks with **unpaired data** under **zero ablation**.

*Figure 8.* SVA unpaired with zero-ablation.

# E. Paired SVA results with Gemma-2 models

We apply the same evaluation framework described in the main text to the Gemma-2 model family to validate that MLP activations form strong bases for constructing sparse circuits across different model architectures. We evaluate both the Gemma-2-2B and Gemma-2-9B models, comparing neuron bases (MLP activations, MLP outputs, residual stream) against SAE bases.

For both models, we evaluate SAEs at two different width scales: for Gemma-2-2B, we use 16k and 65k latent dimensions; for Gemma-2-9B, we use 16k latent dimensions. The smaller width SAEs provide a standard representation, while the larger width SAEs offer an expanded latent space that may capture more fine-grained features. All SAE variants are from Google's Gemma Scope (Lieberum et al., 2024) suite.

### E.1. Gemma-2-2B Results

We present results averaged over the four SVA tasks (`simple`, `rc`, `within_rc`, `nounpp`) for Gemma-2-2B. The findings closely mirror those from the main text: **MLP activations yield significantly sparser circuits** compared to MLP outputs, with circuit sizes reduced by approximately 100x at comparable faithfulness and completeness levels.

With the standard 16k width SAEs, MLP activations substantially close the gap with SAE-based representations while maintaining the computational and interpretability advantages of working directly with neurons. The larger 65k width SAEs offer an expanded latent space with more features, which can potentially capture finer-grained patterns in the model's computation. Even against these higher-capacity SAEs, MLP activations remain competitive and provide substantially sparser circuits than MLP outputs.

### E.2. Gemma-2-9B Results

We similarly evaluate the Gemma-2-9B model with 16k width SAEs on the same SVA benchmark. The results demonstrate that our findings scale to larger models within the Gemma-2 family.

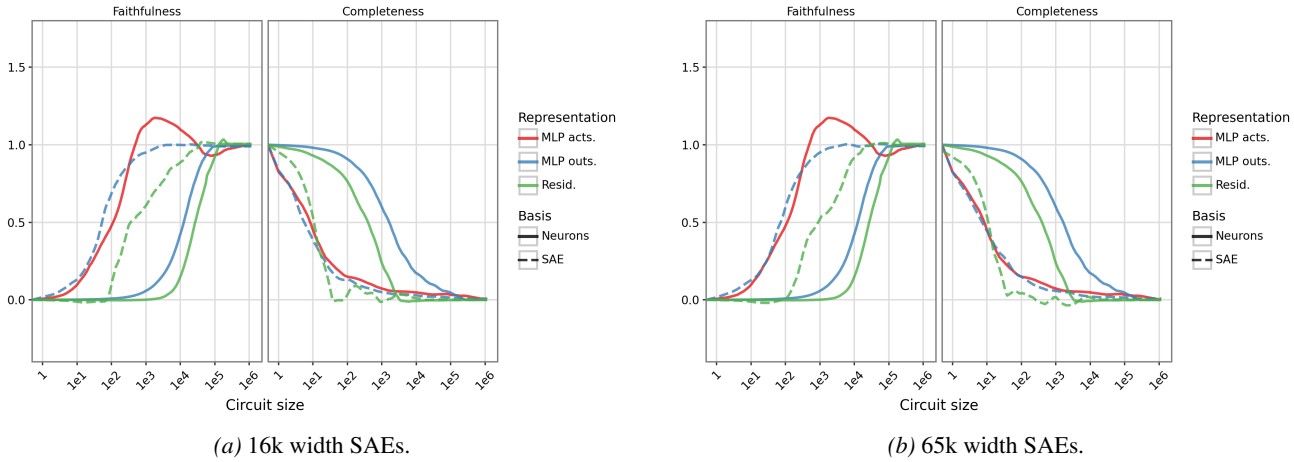

*(a)* 16k width SAEs.

*(b)* 65k width SAEs.

*Figure 9.* Faithfulness and completeness for Gemma-2-2B. MLP activations provide significantly sparser circuits than MLP outputs and approach SAE-based performance across both SAE widths.

**16k Width SAEs.** The Gemma-2-9B model with 16k width SAEs shows consistent patterns with the smaller Gemma-2-2B model. MLP activations continue to provide substantially sparser circuits than MLP outputs while maintaining competitive faithfulness and completeness compared to SAE-based representations.

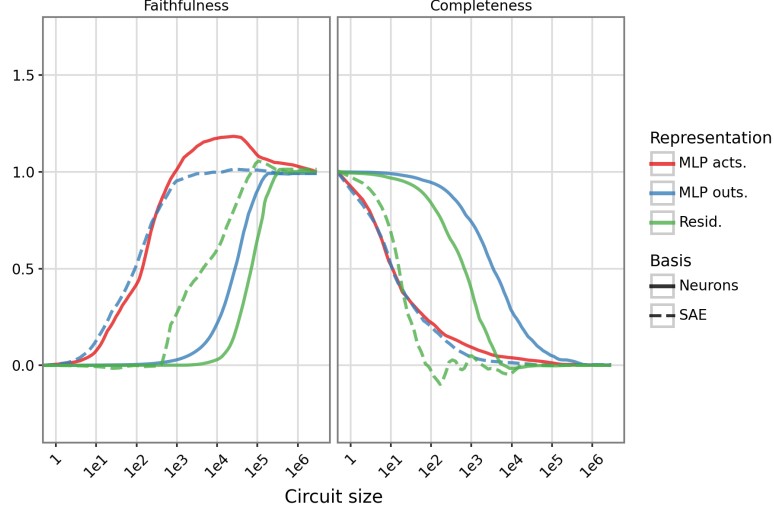

*Figure 10.* Faithfulness and completeness for Gemma-2-9B with 16k width SAEs. MLP activations deliver significantly sparser circuits than MLP outputs and remain competitive with SAE-based approaches.

### E.3. Summary

The results on Gemma-2 models validate our main findings: **MLP activations provide an effective basis for circuit discovery that yields significantly sparser circuits than using MLP outputs**. Additionally, MLP activations yield similar performance to SAE bases, and SAE width does not significantly affect circuit sparsity for SAE-based representations.

## F. MLP activations vs. other neuron bases on paired SVA with IG

We now analyse the statistics of the IG attribution scores on paired SVA. We note some interesting differences in these statistics across different neuron bases in the model, which show why MLP activations may be a more desirable attribution target than alternatives.

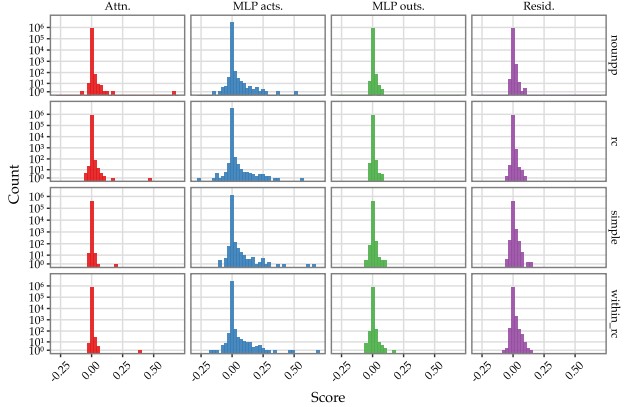

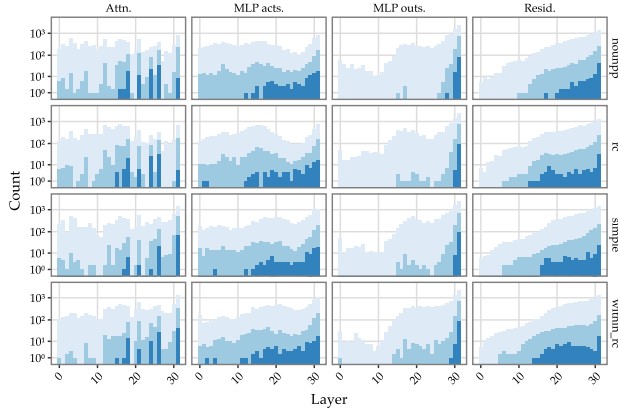

*(a)* Distribution of attribution scores for different choices of representation when applying Integrated Gradients.

*(b)* Distribution of top-attributed neurons by layer for different choices of representation when applying Integrated Gradients. From more to less saturated, the colours represent the top $10^2$, $10^3$, and $10^4$ neurons by absolute attribution.

*Figure 11.* Analysis of attribution patterns **on each of the SVA subsets**.

**MLP activation attributions have greater spread and more outliers.** In Figure 11a, we plot the histogram of attribution scores Attr($v$) for each node $v \in V$ (recall that these scores are averaged over the training set), for each method across all 4 tasks in the histograms below. Compared to other representations in the model, MLP activations have larger spread in both the bulk and tails. Since circuit tracing works by greedily taking the components with highest attribution, larger spread means we need fewer components to reach the same total effect.

**MLP activation attributions are more distributed throughout the model's depth.** We next plot the number of features included in the circuit by layer, in Figure 11b. We plot this for circuits comprising the top 100, 1,000, and 10,000 scoring neurons for each basis. MLP activation scores are more evenly distributed throughout the model's depth than the other bases; MLP outputs are highly concentrated in the last two layers, residual stream scores are somewhat biased towards later layers, and attention outputs are haphazardly distributed with some specific layers having unusually high attribution scores.

### F.1. `nounpp`: MLP activation neuron labels are task-relevant

To better qualitatively understand the circuits that we uncover, we examine all the MLP activation neurons included in a 500-node circuit for the nounpp task. Recall that the nounpp task consists of examples like: *The **secretaries** near the cars* → ***have*** vs. *The **secretaries** near the cars* → ***has***.

We looked at the neuron descriptions and top-activating exemplars for each neuron in the circuit, using Choi et al. (2024). Examining the maximum-activating exemplars alongside the descriptions reveals several task-relevant features at work, related to linguistic features like number, tense, and agreement. We list an unfiltered set of the top neurons grouped by which token they fire on in Table 5; note that token 2 is the noun whose number feature we are alternating, token 3 is the following preposition, and token 5 is the noun right before the model outputs the verb. Expectedly, tokens 2 and 3 generally have middle-layer features encoding noun number while token 5 has very late-layer features encoding output verb number.

We list our analysis of the most important features in Table 4.

| Neuron | Analysis |
|---|---|
| L30/N11158 | Language-specific **singular noun / third-person pronoun** neuron, with positive exemplars being the English inanimate third person pronoun *it* and negative exemplars being various Russian nouns and pronouns (*kotoraja* "who" (fem.), *čelovek* "person", *jakyj* "which" (masc.), etc.) |
| L29/N10537 | Fires language-agnostically **before forms of the word "be"**, e.g. negatively before *siano* "are (3pl.; Italian)", *'m* (1sg.; English), positively before *zijn* "is (3sg.; Dutch)". Some other verbs make it in as well. |
| L30/13476 | Positively fires on **the end of a conjunctive noun phrase** in English, e.g. *Chen and Sandino*, *Bagby and Nimer*, and negatively on **non-English plural verb/noun forms** (e.g. German *werden*, *erreichen*, Ukrainian *vyboriv*) |
| L19/N12056 | Negatively fires on **plural subjects** in English (e.g. *they*, *The most common types...*), perhaps triggered also by definiteness and other markers of grammatical subjecthood; it positively fires less cleanly on what seem to be plural verb forms of the verb for "to be able to" in Spanish and Portuguese. |
| L17/N4140 | Positively fires on **plural nouns across languages** (e.g. English plural acronyms like *ADCs*, *EVs*, etc., Russian *sinimi* (instrumental plural of *sinij* "deep blue")) |

*Table 4.* Analysis of selected neurons and their activation patterns.

*Table 5.* Feature scores for SVA nounpp task.

| Feature | Description | Score |
|---|---|---|
| **Token: 2** | | |
| L19/N12056+ | presence of the token "podem" in contexts dis... | 0.151 |
| L19/N12056− | direct references to objects or concepts, suc... | |
| L17/N4140+ | words indicating qualities or categories such... | 0.112 |
| L17/N4140− | occurrences of the plural suffix -{{ink}}... | |
| L17/N13649+ | activation occurs on tokens indicating collec... | 0.090 |
| L17/N13649− | terms related to categories, roles, or types ... | |
| L19/N725+ | Themes of production processes in the chemica... | 0.058 |
| L19/N725− | the words "pourrait", "pourra", "va", "a", "w... | |
| L23/N5918+ | "each", "one", "every one", and similar terms... | 0.057 |
| L23/N5918− | phrases containing "none of" or "none" leadin... | |
| L25/N529+ | tokens indicating uncertainty or clarificatio... | 0.048 |
| L25/N529− | "voc[U+00EA]", "empresa", "ele", "se", "usu[U... | |
| L25/N8600+ | the term "Times" in various contexts, especia... | 0.048 |
| L25/N8600− | words that introduce lists or examples (e.g.,... | |
| L23/N5186+ | activating tokens signal presence of 2nd pers... | 0.045 |
| L23/N5186− | response words indicating communication in di... | |
| L18/N11717+ | activating verbs like "stands," "calls," "fun... | 0.042 |
| L18/N11717− | pronouns "he", "it", "she", "one" in varied c... | |
| L15/N14016+ | activation occurs on adjectives and some spec... | 0.040 |
| L15/N14016− | tokens indicating a topic or focus, such as "... | |
| L14/N3634+ | the word "{{mass}}" and "{{ability}}"... | 0.038 |

*Table 5.* Feature scores for SVA nounpp task.

| Feature | Description | Score |
|---|---|---|
| L14/N3634− | activating tokens related to specific nouns o... | |
| L12/N10315+ | words indicating importance or essential natu... | 0.036 |
| L12/N10315− | references to specific concepts or categories... | |
| L19/N7387+ | tokens referring to groups, locations, indivi... | −0.036 |
| L19/N7387− | the token "Wins" within various contexts; als... | |
| L20/N7715+ | Tokens "{{there}}" in introductory narrat... | 0.033 |
| L20/N7715− | the word "exist" and variations, often follow... | |
| **Token: 3** | | |
| L15/N14016+ | activation occurs on adjectives and some spec... | 0.056 |
| L15/N14016− | tokens indicating a topic or focus, such as "... | |
| L15/N7096+ | the conjunction "or" following nouns in compa... | 0.050 |
| L15/N7096− | tokens indicating certainty or identity, part... | |
| L17/N4140+ | words indicating qualities or categories such... | 0.039 |
| L17/N4140− | occurrences of the plural suffix -{{ink}}... | |
| L19/N12056+ | presence of the token "podem" in contexts dis... | 0.037 |
| L19/N12056− | direct references to objects or concepts, suc... | |
| **Token: 5** | | |
| L30/N11158+ | occurrences of "it" and "may" as references i... | 0.516 |
| L30/N11158− | the token "[CYR:KA][CYR:O][CYR:TE][CYR:O][CYR... | |
| L29/N10537+ | the tokens "[CYR:TE][CYR:IE][CYR:KA][CYR:ES][... | 0.357 |
| L29/N10537− | activating tokens indicating actions related ... | |
| L30/N13476+ | activation occurs on specific names of schola... | 0.268 |
| L30/N13476− | occurrence of a word associated with actions ... | |
| L31/N5287+ | presence of tokens "[U+00E9]" or "es" indicat... | 0.254 |
| L31/N5287− | the word "intellectually" and the token "and"... | |
| L31/N11075+ | use of third-person plural pronouns ("they", ... | 0.252 |
| L31/N11075− | tokens indicating actions, thoughts, or state... | |
| L21/N8045+ | occurrence of the token {{,}} following e... | 0.213 |
| L21/N8045− | activation occurs after the phrase "go by the... | |
| L31/N3484+ | the token "{{.}}" appearing at the end of... | 0.204 |
| L31/N3484− | the word "Study" or "Who" in contexts related... | |
| L31/N8809+ | presence of specific phrases or tokens indica... | 0.202 |
| L31/N8809− | presence of the token "it", "may", or similar... | |
| L25/N529+ | tokens indicating uncertainty or clarificatio... | 0.202 |
| L25/N529− | "voc[U+00EA]", "empresa", "ele", "se", "usu[U... | |
| L23/N5186+ | activating tokens signal presence of 2nd pers... | 0.197 |
| L23/N5186− | response words indicating communication in di... | |
| L25/N2942+ | Chinese names with the character "[CJK]" and ... | 0.161 |
| L25/N2942− | common English words like "all" and specific ... | |
| L22/N82+ | use of the token "than" in a comparative cont... | 0.160 |
| L22/N82− | pronouns indicating subjects (e.g., "ele", "e... | |
| L26/N3064+ | verbs related to action or occurrence (e.g., ... | 0.159 |
| L26/N3064− | activating tokens mention counts of "sisters"... | |
| L28/N3849+ | Spanish and Italian tokens indicating suggest... | 0.154 |
| L28/N3849− | presence of specific names (e.g. Lamborghini,... | |
| L29/N6353+ | pronouns or subjects like "eles," "se," or "i... | 0.141 |
| L29/N6353− | the presence of "[CYR:PE][CYR:O][CYR:HA][CYR:... | |
| L31/N5187+ | tokens "There," "Here," "there," indicating c... | −0.138 |
| L31/N5187− | the word "G" in the context of train stations... | |
| L28/N292+ | various verb forms (e.g., {{desenv}}, {:. | 0.136 |
| L28/N292− | tokens indicating yearning or desire, e.g., "... | |
| L29/N13065+ | active references to singular entities and ve... | 0.131 |
| L29/N13065− | the word "{{d}}" appearing in a context o... | |
| L30/N5428+ | presence of the phrase "it {{may}}" indic... | 0.124 |
| L30/N5428− | conjunctions and transitions leading to addit... | |
| L29/N1221+ | presence of the word {{que}} or semantic ... | 0.104 |
| L29/N1221− | names of authors and academics (e.g. "Naigles... | |
| L27/N12026+ | presence of first-person singular pronouns ("... | 0.104 |
| L27/N12026− | the token 'interpret', 'cres', '[U+03B2][U+03... | |
| L22/N4588+ | words initiating uncertainty or conditionalit... | 0.098 |
| L22/N4588− | pronouns or relative pronouns introducing add... | |

*Table 5.* Feature scores for SVA nounpp task.

| Feature | Description | Score |
|---|---|---|
| L18/N9714+ | the word "They" or its variations, often refe... | 0.096 |
| L18/N9714− | presence of the token "a" in various contexts... | |
| L31/N5724+ | instances of numbers followed by a comma ({:. | −0.095 |
| L31/N5724− | "it" and "may" | |
| L29/N7925+ | the verbs "escre" and "lle" are highlighted w... | 0.094 |
| L29/N7925− | the token "what" when it initiates questions ... | |
| L29/N11990+ | words with suffixes indicating change or modi... | 0.093 |
| L29/N11990− | activation occurs after plural nouns, specifi... | |
| L19/N12056+ | presence of the token "podem" in contexts dis... | 0.091 |
| L19/N12056− | direct references to objects or concepts, suc... | |
| L19/N725+ | Themes of production processes in the chemica... | 0.089 |
| L19/N725− | the words "pourrait", "pourra", "va", "a", "w... | |
| L24/N12083+ | "that" in various contexts; "all" in contexts... | 0.086 |
| L24/N12083− | second-person pronoun "{{you}}" and third... | |
| L20/N3218+ | references to "inclusion", "evaluation", and ... | −0.085 |
| L20/N3218− | the words "are", "have", "do", "who" used as ... | |
| L31/N6072+ | instances of "then", "also", "can", and "as" ... | 0.081 |
| L31/N6072− | occurrences of "and" or "she" indicating a co... | |
| L26/N11122+ | occurrences of "je" or "io" when referring to... | 0.075 |
| L26/N11122− | activating tokens include "[CYR:ZE][CYR:A][CY... | |
| L28/N1405+ | the specifically targeted term or phrase typi... | 0.075 |
| L28/N1405− | pronouns ("ils", "eux", "elles", "ne") indica... | |
| L23/N7055+ | verbs in the passive voice such as "handelt,"... | 0.075 |
| L23/N7055− | references the word "there" in contexts discu... | |
| L31/N12358+ | the token {{:}} used before a list of per... | 0.074 |
| L31/N12358− | activating tokens from information statements... | |
| L30/N10462+ | the word "who," "may," and "that" in contexts... | 0.073 |
| L30/N10462− | specific word forms derived from root words i... | |
| L28/N5914+ | the conjunction "and" when listing features, ... | 0.073 |
| L28/N5914− | presence of tokens indicating existence or ne... | |
| L31/N5861+ | references to self-identity (e.g., "this mode... | −0.073 |
| L31/N5861− | tokens like "including," "which," and "for" t... | |
| L20/N8563+ | "someone {{who}}"/"someone {{that}}"/... | −0.072 |
| L20/N8563− | use of {{will}} to indicate future action... | |
| L31/N437+ | instances of "to" indicating direction of act... | 0.072 |
| L31/N437− | token references to objects or female pronoun... | |
| L20/N7715+ | Tokens "{{there}}" in introductory narrat... | 0.072 |
| L20/N7715− | the word "exist" and variations, often follow... | |
| L30/N9634+ | presence of conjunctions like "and" and "may,... | 0.067 |
| L30/N9634− | respond/engagement tokens that imply user int... | |
| L30/N11949+ | activating occurrences of the token "ch{{an... | 0.064 |
| L30/N11949− | the word "there" positioned in contexts discu... | |
| L26/N9709+ | Activation occurs on the pronoun "I" or its v... | 0.064 |
| L26/N9709− | Response structure with confirmation or elabo... | |
| L30/N6601+ | the pronoun "it" in contexts discussing capab... | −0.063 |
| L30/N6601− | the phrase "have {{to}}" in various conte... | |
| L31/N13933+ | tokens with complex or specialized vocabulary... | 0.063 |
| L31/N13933− | reliance on context phrases like "ENEMY" or "... | |
| L28/N10371+ | activation on specific tokens (e.g. "eu", "c"... | 0.062 |
| L28/N10371− | the presence of specific nouns or keywords (:. | |
| L28/N12606+ | the word "cliente" when referring to dissatis... | 0.062 |
| L28/N12606− | tokens "Ich", "eu", "que" indicating self-ref... | |
| L28/N10596+ | activating verb forms and words indicating ne... | 0.059 |
| L28/N10596− | references to people and entities with "who",... | |
| L30/N2638+ | the infinitive form "to" and "will" in contex... | 0.058 |
| L30/N2638− | the token "{{tw}}" | |
| L30/N11956+ | the word "hier" following a suggestion or rec... | 0.051 |
| L30/N11956− | the token "altre", "delle", "nelle", "de", an... | |
| L25/N8600+ | the term "Times" in various contexts, especia... | 0.051 |
| L25/N8600− | words that introduce lists or examples (e.g.,... | |
| L31/N11269+ | contextually relevant or specific roles/nouns... | 0.048 |
| L31/N11269− | The word "as" when used to introduce explanat... | |

*Table 5.* Feature scores for SVA nounpp task.

| Feature | Description | Score |
|---|---|---|
| L27/N9333+ | tokens referring to groups or entities (e.g.,... | 0.048 |
| L27/N9333− | the token "and" or "will" appearing in contex... | |
| L31/N1834+ | the token "applications" and the phrase "on" ... | 0.047 |
| L31/N1834− | conjunctions, especially "but," and certain c... | |
| L31/N13110+ | mentions of specific items, particularly thos... | 0.046 |
| L31/N13110− | the token "is" before a definition or descrip... | |
| L30/N11344+ | references to "you" and "I" indicating a dire... | −0.046 |
| L30/N11344− | use of the word "do" in the context of respon... | |
| L26/N280+ | words leading to engagement or a responsive p... | −0.045 |
| L26/N280− | activation caused by specific names, such as ... | |
| L31/N10088+ | references to "they," "you," and "we" to deno... | 0.045 |
| L31/N10088− | presence of pronouns indicating second and fi... | |
| L30/N6976+ | activation occurs on tokens that include punc... | 0.043 |
| L30/N6976− | presence of article "a" or "les" or "des" lea... | |
| L24/N7205+ | Tokens that show reference or indication: {:. | 0.042 |
| L24/N7205− | occurrences of the word "there" in diverse co... | |
| L29/N7067+ | words like {{we}}, {{also}}, {{them... | 0.041 |
| L29/N7067− | singular first-person pronouns ({{I}}) an... | |
| L29/N592+ | tokens indicating a theme, situation, or absc... | −0.041 |
| L29/N592− | presence of specific term or noun (e.g. "[U+B... | |
| L29/N3626+ | references to durations and timeframes (e.g. ... | −0.041 |
| L29/N3626− | activation occurs on question words {{que}... | |
| L18/N11739+ | the word "they," often referencing a group me... | 0.041 |
| L18/N11739− | Presence of the pronoun "I" preceding activat... | |
| L27/N5290+ | words expressing certainty, opinion, or respo... | −0.040 |
| L27/N5290− | the appearance of "there" (English), "robotiq... | |
| L21/N7761+ | tokens introducing conditions or comparisons ... | 0.039 |
| L21/N7761− | activation occurs on the words "cards," "ther... | |
| L18/N6570+ | tokens indicating identity or action (I, its,... | 0.039 |
| L18/N6570− | references to personal affliction, emotional ... | |
| L30/N6906+ | affirmative phrases like "{{vil gerne}}" ... | 0.038 |
| L30/N6906− | the relative pronoun "qui" referring to peopl... | |
| L30/N2697+ | the token "this" or "that" referring to a nou... | 0.038 |
| L30/N2697− | conjunctions or terms indicating additional i... | |
| L29/N2062+ | specific words related to scientific or techn... | 0.037 |
| L29/N2062− | usage of "this" and "this performance" in con... | |
| L31/N2420+ | presence of a second-person pronoun "{{you.. | 0.037 |
| L31/N2420− | activation occurs on modifiers that precede a... | |
| L28/N1488+ | occurrence of "car", "short" and "interesting... | 0.036 |
| L28/N1488− | the token "must", plural "lists", the token "... | |
| L18/N13748+ | tokens "are" and "them" in various contexts. | −0.036 |
| L18/N13748− | the presence of specific names, particularly ... | |
| L29/N3534+ | activating phrases indicating actions or stat... | 0.036 |
| L29/N3534− | phrases that include "you {{could}} ...",... | |
| L31/N11276+ | the token "{{I}}" at the start of a perso... | 0.035 |
| L31/N11276− | occurrences of "{{(I)}}" and "{{I}}" o... | |
| L31/N9804+ | the presence of the absolute form of certain ... | 0.035 |
| L31/N9804− | occurrences of self-referential or context-or... | |
| L22/N10532+ | Hindi words with specific verb conjugations, ... | 0.035 |
| L22/N10532− | the word "are" appearing to indicate a defini... | |
| L30/N3440+ | text tokens that refer to personal identity o... | 0.035 |
| L30/N3440− | second-person pronouns, specifically {{you.. | |
| L30/N6083+ | noun "the" preceded by context of service, st... | 0.034 |
| L30/N6083− | activations on conversational and informal ph... | |
| L29/N1654+ | single word "it" or "may" following a request... | 0.034 |
| L29/N1654− | words indicating roles or functions ({{[CYR... | |
| L31/N6614+ | the word "[CJK]" prominently activating befor... | 0.033 |
| L31/N6614− | the token "work" and mentions of time ("{{... | |

# G. More details on circuit tracing

### G.1. Filtered neurons

Additionally, we manually filtered out a few neurons which we found were present in the circuit we traced, across every dataset and at many token positions. These neurons are always activated and thus do not seem to provide useful task-specific information when included in circuit analysis. The filtered neurons are: L23/N306, L20/N3972, L18/N7417, L16/N1241, L13/N4208, L11/N11321, L10/N11570, L9/N4255, L7/N6673, L6/N5866, L5/N7012, L2/N4786.

### G.2. Prompt template

This is the exact prompt template we use for Llama 3.1 8B Instruct on the capitals task in §6:

```
<|begin_of_text|><|start_header_id|>system<|end_header_id|>
<|eot_id|><|start_header_id|>user<|end_header_id|>

What is the capital of the state containing Dallas?
<|eot_id|><|start_header_id|>assistant<|end_header_id|>

Answer:
```

All other tasks use a similar template, with the assistant prefilled to say "Answer:" and only the prompt being changed depending on the task.

# H. Additional case studies

**Case studies.** First, we replicate two more case studies introduced in Lindsey et al. (2025) and Ameisen et al. (2025):

- Addition problems
- Multilingual antonym prediction

Additionally, we investigate **user modeling**, which prior circuit tracing work has not studied. These add additional evidence to §6 that our circuit tracing method finds useful structure across a variety of datasets.

We follow the same exact methodology from §6.

**Finding interesting neurons with labelled data.** For each of our datasets, there are natural example-level properties that should have corresponding model-internal referents. For example, in our addition dataset, we expect to find features that encode the answer modulo 10 (since this is feature was found in the CLT circuit from Lindsey et al.). To identify nodes that encode these properties, we introduce a scoring function: given example-level labels of the property of interest, we score a node $v$ based on how well its attribution score separates classes of labels.

Concretely, assume we have a labelling function $f$ that maps $x \sim \mathcal{D}$ to a categorical label $a \in A$ (for example, the answer mod 10). First, we select a specific class $a \in A$ (e.g. the answer modulo 10 is 7). We split the dataset into positive and negative examples for $a$:

$$\mathcal{D}^+ = \{x \in \mathcal{D} : f(x) = a\} \tag{43}$$

$$\mathcal{D}^- = \{x \in \mathcal{D} : f(x) \neq a\} \tag{44}$$

Then, for a node $v$, we compute its attribution score for each of the positive and negative examples, and compute the AUROC over the dataset:

$$\text{AUROC}(v) = \mathbb{P}_{x^+ \sim \mathcal{D}^+, x^- \sim \mathcal{D}^-}[\text{Attr}_v(x^+) > \text{Attr}_v(x^-)] \tag{45}$$

This tells us how well this node's attribution score correlates with the specific label $a$ of interest. We apply this procedure for each $a \in A$ to find interesting neurons in a supervised manner, reporting all neurons in the circuit that have large AUROC. In each of our case studies, we will analyze these attribute-sensitive neurons, looking at their top-activating exemplars, steering effects, and so on.

### H.1. Addition

Ameisen et al. (2025) uncover CLT features underlying simple addition problems in Claude 3.5 Haiku, as well as in an 18-layer toy model. This task is interesting because of its complexity: there are diverse task-specific features such as tracking the ones digit and tens digit of the answer.

We consider a similar task (rephrased into an instruction format) for Llama 3.1 8B Instruct and succeed in uncovering the same types of features in the MLP neuron basis via neuron-level circuit tracing. In particular, we examine the following categories of features:

*Table 6.* Arithmetic features.

| Feature | Expression | Replicated? | Example neuron |
|---------|-----------|-------------|----------------|
| Ones digit (sum) | $(x + y) \bmod 10$ | Yes | L21/N10677 |
| Mod-$n$ (sum) | $(x + y) \bmod n$ | New | L21/N9178 ($n = 2$) |
| Tens digit (sum) | $\lfloor (x + y)/10 \rfloor \bmod 10$ | Yes | L28/N9549 |

For each feature category, we label the dataset examples (e.g. under mod-10 labels, the label of the example below is 3) and compute AUROCs for each neuron over the dataset based on the attribution score. We then manually examine neurons with AUROC close to 0 or 1, since these are highly predictive of the feature of interest.[8]

Notably, prior work has already found the same features in the MLP neuron basis for this task (Nikankin et al., 2025); our goal is to show that these features meaningfully contribute to the model's output and can be understood as parts of the circuit underlying that computation.

**Dataset.** We construct a dataset of addition problems with operands in the range $[0, 99]$, resulting in 10,000 examples like the following:

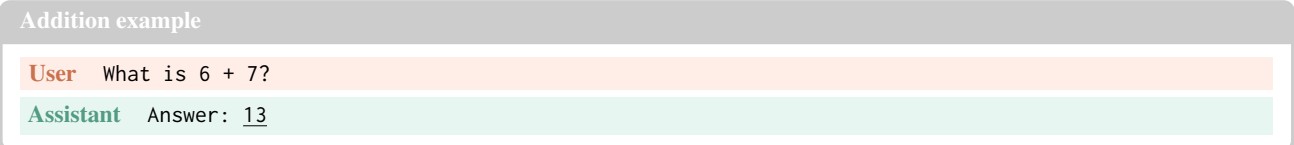

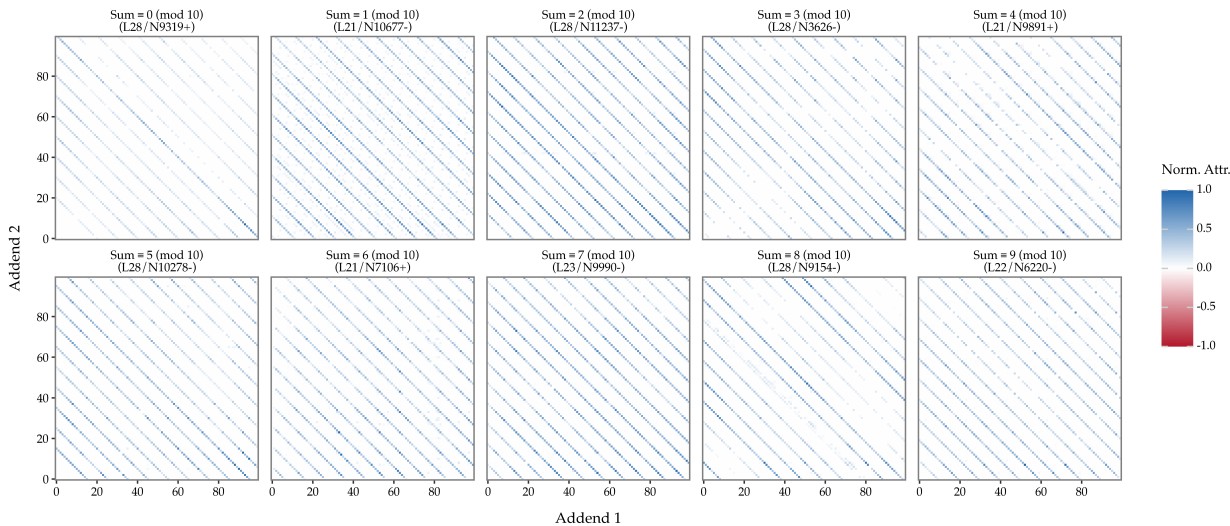

*Figure 12.* Attribution heatmaps of the top-AUROC neuron for mod-10 features. The axes are the addends in the addition prompt.

---

[8]Recall that an AUROC of 0.5 corresponds to random performance; AUROCs far from 0.5 mean the neuron gives signal about the feature of interest (where AUROC $< 0.5$ means the negated activations are a good classifier). Therefore, we sort AUROCS by the absolute distance from 0.5.

### H.1.1. MOD-10 (ONES DIGIT) NEURONS

We find high-AUROC neurons for each outcome of the answer modulo 10. For all of the ten outcomes, we successfully find neurons with AUROC $\geq 0.9$ or AUROC $\leq 0.1$; in some cases, we find near-perfect AUROCs.

We show the per-example attribution scores over the dataset of the top-AUROC neuron for each outcome in Figure 12; the $x$-axis is the first addend, the $y$-axis is the second addend, and the colour of the cell is the attribution score when the model is asked to compute the sum of the two addends. We sum over the token axis to get a single attribution score for each example.

The attribution scores reveal a clean diagonal pattern, meaning that these neurons only play a causal role in the output when the sum is congruent to some value modulo 10. This plot replicates the one in Ameisen et al. (2025); see the "sum = _5" feature in their CLT graph.

We also show all neurons which achieve AUROC $\geq 0.8$ or AUROC $\leq 0.2$ in Table 7. Positively-attributed neurons (AUROC $> 0.5$) are significantly more common.

We note that the automatic neuron descriptions do not indicate their mod-10 role (instead describing a more general context the neuron fires in, such as numbers or dates). Some do note the role indirectly however, e.g. L28/N10436−, which is highly correlated with the ones digit being 7, has the description "year format (e.g., {{201}}7 or {{179}}7) in contexts discussing US presidents or historical timelines".

*Table 7.* Feature scores for sum mod 10 task.

| Feature | Description | AUROC | In-class | Out-of-class |
|---|---|---|---|---|
| **Target: 0** | | | | |
| L22/N12637+ | +numerical values and calculations | 0.935 | 1.33% | 0.00% |
| L28/N9319+ | +text containing numerical data or mathema... | 0.955 | 1.13% | 0.00% |
| L19/N12494+ | +structural elements like year ranges or n... | 0.906 | 0.88% | 0.11% |
| L26/N10556− | −tokens related to numbers and their relat... | 0.809 | −0.70% | −1.13% |
| L25/N9102− | −isolated commas, periods, and numeric rep... | 0.883 | 0.60% | 0.04% |
| **Target: 1** | | | | |
| L21/N10677− | −the conjunction "and" preceding lists or ... | 0.999 | 2.40% | 0.04% |
| L25/N13847+ | +activation on numerical comparisons or re... | 0.971 | 2.10% | 0.00% |
| L19/N12494+ | +structural elements like year ranges or n... | 0.869 | 0.74% | 0.12% |
| L30/N936+ | +phrases starting with "here are" and spec... | 0.194 | −0.70% | −0.23% |
| L27/N11932+ | +tokens referring to the application or us... | 0.145 | −0.63% | −0.36% |
| **Target: 2** | | | | |
| L28/N11237− | −activation occurs before a token indicati... | 0.998 | 3.67% | 0.00% |
| L31/N5580− | −ID numbers or numerical results from calc... | 0.129 | −1.77% | −0.00% |
| L21/N10677+ | +tokens representing numerical intervals o... | 0.187 | −0.13% | −0.02% |
| **Target: 3** | | | | |
| L28/N3626− | −direct reference to a date (e.g., "25", "... | 0.952 | 1.24% | 0.00% |
| L24/N5960− | −enumerated list items (i, ii, iii) in var... | 0.829 | 1.16% | −0.00% |
| L29/N11270− | −The activation is caused by the presence ... | 0.199 | −1.16% | −0.94% |
| **Target: 4** | | | | |
| L21/N9891+ | +phrases indicating numerical or statistic... | 0.939 | 0.80% | 0.01% |
| L23/N6221+ | +the token {{c}} occurring in a list o... | 0.807 | 0.64% | −0.00% |
| **Target: 5** | | | | |
| L28/N10278− | −responding to numerical information or re... | 0.995 | 1.93% | 0.00% |
| L24/N4047− | −activation occurs on the token {{-}} ... | 0.930 | 1.25% | −0.00% |
| L21/N9042+ | +numeric values and certain conjunctions i... | 0.901 | 0.77% | 0.00% |
| L31/N9428+ | +activating closing braces "{{" | 0.946 | 0.50% | 0.04% |
| L31/N9428− | −numbers and temperatures followed by comm... | 0.939 | 0.00% | −0.61% |
| **Target: 6** | | | | |
| L21/N7106+ | +the conjunction "and" in lists or combina... | 0.981 | 1.58% | 0.01% |
| L23/N10338− | −activation occurs on numeric sequences an... | 0.962 | 1.20% | 0.00% |
| L31/N9428− | −numbers and temperatures followed by comm... | 0.199 | −1.17% | −0.48% |
| L20/N3964− | −academic or formal content, dates, calcul... | 0.912 | 1.09% | 0.21% |

*Table 7.* Feature scores for sum mod 10 task.

| Feature | Description | AUROC | In-class | Out-of-class |
|---|---|---|---|---|
| L28/N9504− | ⁻tokens representing years, specifically ⁚. | 0.919 | 0.93% | 0.00% |
| L30/N10778− | ⁻the token {{[U+2013]}} in contexts in... | 0.873 | 0.86% | 0.05% |
| L24/N12344+ | ⁺context requiring a year, a percentage, a... | 0.968 | 0.75% | 0.00% |
| L24/N11719− | ⁻tokens like "and," "word," "term," and "p... | 0.175 | −0.72% | −0.46% |
| L31/N4503+ | ⁺activation around numeric values with com... | 0.811 | 0.55% | 0.07% |
| L27/N5097− | ⁻the tokens "{{-h}}" and "{{b}}" i... | 0.155 | −0.42% | 0.00% |
| L28/N10436+ | ⁺References to days of the week, particula... | 0.808 | 0.36% | 0.12% |
| **Target: 7** | | | | |
| L28/N10436− | ⁻year format (e.g., {{201}}7 or {{17... | 0.990 | 1.70% | 0.00% |
| L23/N9990− | ⁻mentions of numerical data, dollar signs,... | 0.997 | 1.64% | 0.00% |
| L30/N8574+ | ⁺the token {{,}} followed by numeric r... | 0.190 | −1.37% | −0.02% |
| L20/N3964− | ⁻academic or formal content, dates, calcul... | 0.924 | 1.19% | 0.20% |
| L28/N9154+ | ⁺references to years, dates, or historical... | 0.818 | 0.31% | 0.02% |
| **Target: 8** | | | | |
| L28/N9154− | ⁻activation on pivotal years {{161}}, ... | 0.945 | 1.20% | 0.00% |
| L19/N10804− | ⁻monetary values, dates, and specific nume... | 0.902 | 0.73% | 0.09% |
| **Target: 9** | | | | |
| L21/N9178− | ⁻the token "and" linking items in a list, ... | 0.812 | 1.04% | 0.37% |
| L22/N6220− | ⁻specific numerical terms, calculations, o... | 0.977 | 1.00% | 0.00% |
| L30/N7476− | ⁻the token [U+201C]Every[U+201D] in the co... | 0.191 | −0.89% | −0.01% |
| L25/N5830+ | ⁺tokens specifically representing numerica... | 0.808 | 0.79% | 0.00% |
| L31/N2488− | ⁻dates or days of the week, specific to se... | 0.844 | 0.73% | 0.12% |
| L19/N10804− | ⁻monetary values, dates, and specific nume... | 0.868 | 0.65% | 0.10% |
| L21/N1333− | ⁻terms or phrases with deliberate spelling... | 0.847 | 0.44% | 0.01% |

### H.1.2. MOD-$n$ NEURONS

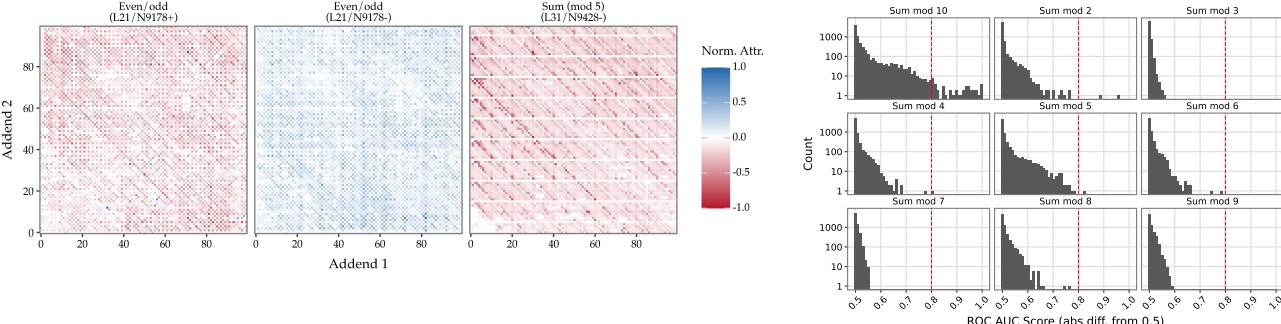

*(a)* Attribution heatmaps of the three neurons with AUROC $\geq 0.8$ or $\leq 0.2$ for mod-$n$ features where $n \neq 10$. The axes are the addends in the addition prompt.

*(b)* Distribution of maximum AUROCs for each mod-$n$ feature (note the logarithmic $y$-axis).

*Figure 13.* Mod-$n$ neuron search results.

As a robustness check, we also look for mod-$n$ features for values of $n$ other than 10; this checks for false positive noise in our analysis; e.g. we don't generally expect to find mod-3 neurons for base-10 addition.

We repeat the procedure above for each $n \in \{2, 3, \ldots, 9\}$; we largely do not find any neurons with high AUROCs for any other value of $n$, except for $n = 2$ (a single neuron with two polarities which strongly promotes odd sums when positive) and $n = 5$ (a single neuron in the final layer which negatively affects the output when the sum is *not* divisible by 5). We note that prior work found evidence of subspaces tracking the units digit mod $2, 5, 10$ for in-context addition (Hu et al., 2025).

To visualize the overall distribution of mod-$n$ neurons, we plot the distribution of AUROCs for each value of $n$ in Figure 13b (note the logarithmic $y$-axis).

For $n$ that are co-prime with 10, the AUROCs stays close to random (between 0.4 and 0.6). For $n = 4, 6, 8$, AUROCs are larger but still generally below 0.8 (The singular exception for mod-4 is actually the mod-2 neuron above). Only mod-10 shows more than a couple neurons with AUROCs greater than 0.8 or less than 0.2.

### H.1.3. TENS DIGIT NEURONS

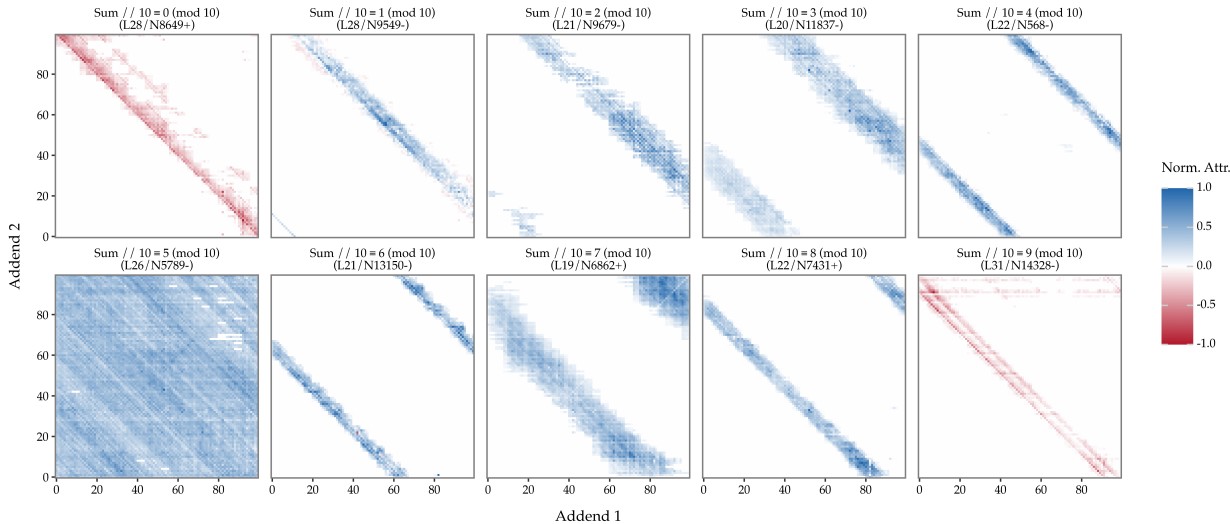

*Figure 14.* Attribution heatmaps of the top-AUROC neuron for tens digit features. The axes are the addends in the addition prompt.

Finally, we look for tens-digit neurons. We find a substantial number of neurons with high AUROCs for each outcome, but their attribution matrices are generally noisier than the ones-digit neurons. Rather than computing the tens digit directly, these neurons may instead be approximating the overall sum (another type of feature that was proposed in the original study by Ameisen et al., 2025).

We show the attribution matrices of the top neuron by AUROC for each outcome in Figure 14.

We also list all neurons which achieve AUROC $\geq 0.8$ or AUROC $\leq 0.2$ on these classes in Table 8.

As before, descriptions are often not specific, except for a few neurons with decade-related descriptions (e.g. L24/N8034+, which activates when the tens digit is 6 and has the description "historical context, notably related to Lyndon B. Johnson and major events...").

*Table 8.* Feature scores for sum tens digit task.

| Feature | Description | AUROC | In-class | Out-of-class |
|---|---|---|---|---|
| **Target: 0** | | | | |
| L30/N10649− | ⁻tokens indicating numeric values, especia... | 0.152 | −1.91% | −0.00% |
| L29/N4814− | ⁻occurrences of numeric tokens, specifical... | 0.805 | 0.92% | 0.01% |
| L29/N13695− | ⁻tokens indicating numerical results or eq... | 0.170 | −0.90% | −0.02% |
| L28/N8649+ | ⁺tokens "one", "HAL" followed by "9000", a... | 0.081 | −0.76% | −0.03% |
| L25/N5270− | ⁻the token {{ }} indicating a range or... | 0.161 | −0.75% | −0.23% |
| L31/N1290− | ⁻the word "in" when followed by a year for... | 0.867 | 0.69% | 0.19% |
| L28/N3343+ | ⁺the activation occurs on specific context... | 0.168 | −0.58% | −0.26% |
| L31/N2718− | ⁻references to specific numerical values o... | 0.828 | 0.55% | 0.02% |
| L27/N3907+ | ⁺references to "into" labor or conditions ... | 0.830 | 0.49% | 0.00% |
| L27/N10962− | ⁻activation occurs after a significant num... | 0.884 | 0.42% | 0.01% |
| L29/N8746+ | ⁺mathematical operations and numerical res... | 0.830 | 0.38% | 0.01% |
| L31/N5910− | ⁻DOI and tax identification number formats... | 0.839 | 0.37% | −0.00% |
| L31/N9987− | ⁻activation occurs on date-related or age-... | 0.197 | −0.35% | −0.05% |
| **Target: 1** | | | | |
| L28/N9549− | ⁻activation occurs when a number or calcul... | 0.891 | 1.34% | 0.01% |

*Table 8.* Feature scores for sum tens digit task.

| Feature | Description | AUROC | In-class | Out-of-class |
|---------|-------------|-------|----------|--------------|
| L25/N5270− | ⁻the token {{ }} indicating a range or... | 0.165 | −0.93% | −0.21% |
| L31/N1290− | ⁻the word "in" when followed by a year for... | 0.891 | 0.70% | 0.19% |
| L27/N3024+ | ⁺presence of colons or parentheses amidst ... | 0.801 | 0.29% | 0.01% |
| **Target: 2** | | | | |
| L30/N5577+ | ⁺activation occurs on specific numbers tha... | 0.194 | −1.43% | −0.00% |
| L19/N5338− | ⁻specific numerical values or tokens indic... | 0.862 | 1.11% | 0.29% |
| L21/N9679− | ⁻the token "and{{ }}" when it appears ... | 0.864 | 0.57% | 0.05% |
| L31/N9549− | ⁻date tokens in the form of {{MM}}, {... | 0.198 | 0.24% | 0.58% |
| **Target: 3** | | | | |
| L20/N11837− | ⁻numbers or statistics; specific percentag... | 0.932 | 1.05% | 0.14% |
| L27/N3934− | ⁻searches or responses in languages other ... | 0.199 | −0.61% | −0.35% |
| **Target: 4** | | | | |
| L22/N568− | ⁻the token "{{ }}" when it acts as a p... | 0.965 | 1.65% | 0.00% |
| L30/N8980− | ⁻presence of numbers, especially those fol... | 0.189 | −1.12% | −0.01% |
| L20/N11837− | ⁻numbers or statistics; specific percentag... | 0.926 | 1.02% | 0.14% |
| **Target: 5** | | | | |
| L19/N13348− | ⁻presence of parentheses or commas in conj... | 0.811 | 0.65% | 0.09% |
| L26/N5789− | ⁻highlighted numbers within the context of... | 0.149 | 0.40% | 0.59% |
| **Target: 6** | | | | |
| L26/N8050− | ⁻the opening token of parentheses {{(}∶. | 0.801 | 0.73% | 0.00% |
| L24/N8034+ | ⁺historical context, notably related to Ly... | 0.882 | 0.70% | 0.02% |
| L18/N7477− | ⁻mentions of numerical details, particular... | 0.161 | −0.62% | −0.18% |
| L21/N13150− | ⁻sequences indicating pagination or numeri... | 0.946 | 0.61% | 0.00% |
| L27/N6959− | ⁻references to specific dates or years in ... | 0.831 | 0.39% | 0.00% |
| L23/N12230+ | ⁺specific instances of "the" before signif... | 0.819 | 0.33% | 0.00% |
| **Target: 7** | | | | |
| L31/N4769+ | ⁺occurrences of the token "{{'" or "{{... | 0.136 | −1.71% | −0.22% |
| L19/N6862+ | ⁺presence of page ranges in publication re... | 0.884 | 1.27% | 0.27% |
| L30/N6208− | ⁻year references in a legal or historical ... | 0.193 | −1.07% | −0.03% |
| L19/N5338− | ⁻specific numerical values or tokens indic... | 0.829 | 0.99% | 0.30% |
| L23/N12607− | ⁻mentions of notable films, trials, and ev... | 0.806 | 0.77% | 0.03% |
| L28/N4868+ | ⁺the token "{{ }}" appears in the cont... | 0.180 | −0.49% | −0.01% |
| L30/N6642− | ⁻a mention of a numeric value or quantity ... | 0.195 | −0.46% | 0.00% |
| L24/N11746+ | ⁺tokens representing dates or timestamps, ... | 0.800 | 0.45% | 0.15% |
| L27/N13853− | ⁻syntax errors in code snippets (e.g. uncl... | 0.854 | 0.44% | 0.14% |
| **Target: 8** | | | | |
| L19/N6862+ | ⁺presence of page ranges in publication re... | 0.941 | 1.59% | 0.24% |
| L22/N7431+ | ⁺tokens indicating page ranges, specifical... | 0.987 | 0.97% | 0.01% |
| L29/N7280+ | ⁺Activation occurs on tokens representing ... | 0.894 | 0.91% | 0.00% |
| L30/N1231− | ⁻the character "{" before numbers or rela... | 0.158 | −0.90% | −0.03% |
| L20/N9739+ | ⁺mathematical or statistical values and ca... | 0.865 | 0.86% | 0.11% |
| L24/N9147+ | ⁺mentions of movies or television shows, p... | 0.852 | 0.63% | 0.00% |
| L31/N7221+ | ⁺mentions of historical events or figures ... | 0.824 | 0.59% | 0.25% |
| L30/N9625− | ⁻references to years or numerical informat... | 0.157 | −0.50% | −0.00% |
| L27/N13951+ | ⁺tokens representing years, conjunctions, ... | 0.807 | 0.25% | 0.01% |
| **Target: 9** | | | | |
| L28/N6838− | ⁻years and dates (e.g., "182", "197", "584... | 0.851 | 2.36% | 1.40% |
| L29/N9469− | ⁻tokens related to numbers or numerical da... | 0.811 | 1.61% | 1.11% |
| L30/N7201+ | ⁺word "los" in various contexts and "proba... | 0.116 | −1.09% | −0.01% |
| L31/N4769− | ⁻presence of numeric data or statistical f... | 0.880 | 0.89% | 0.08% |
| L31/N14328− | ⁻the word "all" in the context of file pro... | 0.055 | −0.84% | −0.03% |
| L20/N9739+ | ⁺mathematical or statistical values and ca... | 0.816 | 0.81% | 0.12% |
| L31/N7221+ | ⁺mentions of historical events or figures ... | 0.917 | 0.74% | 0.23% |
| L26/N3988− | ⁻mentions of specific numbers | 0.921 | 0.63% | 0.00% |
| L28/N13880+ | ⁺numeric values following a dot {{.}},... | 0.916 | 0.59% | 0.00% |

*Table 8.* Feature scores for sum tens digit task.

| Feature | Description | AUROC | In-class | Out-of-class |
|---------|-------------|-------|----------|--------------|
| L25/N6283+ | $^{+}$presence of the preposition "by," tempora... | 0.862 | 0.50% | 0.04% |
| L25/N1402+ | $^{+}$words representing numbers (e.g., "hundre... | 0.898 | 0.42% | 0.02% |
| L22/N3193− | $^{-}$Page numbers or numerical references foll... | 0.837 | 0.33% | 0.00% |

## H.2. Multilingual antonyms

As a final replication of prior CLT circuits results, we investigate multilingual circuits for finding antonyms in Llama 3.1 8B Instruct, replicating the "multilingual circuits" case study from Lindsey et al. (2025). In this task, the model is asked to say the antonym of a given word, with prompts given in several languages.

**Dataset.** We construct a multilingual dataset in which the model is asked to return the antonym of a given word. The prompts are constructed over 9 languages (English, Chinese, French, German, Spanish, Italian, Russian, Hindi, Arabic) and 6 concepts ("big", "small", "fast", "slow", "hot", "cold"), resulting in 54 prompts.

---

**Multilingual antonyms**

**User**   What is the opposite of big?

**Assistant**   Answer: _small_

---

**Circuit tracing.** We perform automatic circuit tracing with a threshold of $\tau = 0.005$ (as usual) on each of the 54 examples in the dataset.

We look for neurons encoding three kinds of features:

*Table 9.* Multilingual antonym features found in Claude Sonnet 3.5.

| Feature | Description | Replicated? | Example neuron |
|---------|-------------|-------------|----------------|
| Language | the language of the prompt | Yes | L31/N8258 (output text in Chinese) |
| Concept | the language-independent meaning of the word being asked about (e.g. "hot") | Yes | L16/N1694 (hot) |
| Attribute | the semantic axis along which the word and its antonym belong (e.g. "temperature") | Yes | L14/N13885 (temperature) |

We do not find any single feature that universally encodes the "antonym" relation (unlike e.g. the state capitals task, where we found a highly important state capital neuron that is active on every prompt), but we do find multiple neurons encoding language, concept, and attribute features.

**Analysis.** We compute the AUROCs of each neuron for each of the three features (language, concept, and attribute) and plot the histograms of the maximum AUROCs for each neuron in Figure 15a. We find hundreds of neurons strongly encoding language information, and tens of neurons encoding concept and attribute information. We set thresholds for further investigation: $\mathrm{AUROC} \geq 0.9, \leq 0.1$.

We also plot the layers at which these filtered neurons are located in the model in Figure 15b.

Language neurons are distributed throughout the model's depth with a large peak at the final layers, concept neurons are in the early and middle layers, and attribute neurons only arise in the middle layers. We study the top neurons for each feature type below, finding relevant descriptions in each case.

### H.2.1. LANGUAGE

The language-specific neurons are numerous and often have relevant descriptions, such as L31/N4787− which has the description "activation on Arabic grammatical forms and prefixes...within religious or formal Arabic context." In general,

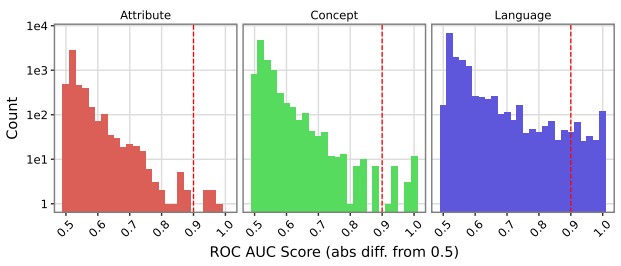

*(a)* Distribution of maximum AUROCs for each feature.

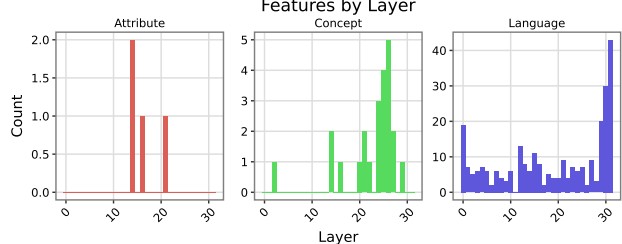

*(b)* Distribution of layers at which filtered neurons are located.

*Figure 15.* Multilingual antonyms task results.

these neurons are distributed throughout the model and may be responsible for various subcategories of language-specific processing (e.g. producing words of a specific part-of-speech in some language). We do not find any single neuron that controls the output language by itself. We show the top 20 neurons by AUROC for the language feature in the table below.

*Table 10.* Feature scores for language task.

| Feature | Description | AUROC | In-class | Out-of-class |
|---|---|---|---|---|
| **Target: ar** | | | | |
| L31/N4787− | ⁻activation on Arabic grammatical forms an... | 1.000 | 9.42% | 0.01% |
| L31/N280+ | ⁺tokens associated with question words and... | 1.000 | 4.51% | 0.00% |
| L27/N13732− | ⁻Tokens highlighting relationships (e.g., ... | 1.000 | 3.27% | 0.00% |
| L31/N12172− | ⁻activation occurs after Arabic tokens. | 1.000 | 3.16% | 0.00% |
| L29/N2205− | ⁻The neuron activates on the presence of c... | 0.045 | −3.00% | −0.54% |
| L24/N13320− | ⁻the words "[AR:ALEF][AR:LAM][AR:TEH][AR:Y... | 1.000 | 2.46% | 0.00% |
| L31/N747− | ⁻Korean language grammatical structures or... | 0.000 | −2.44% | −0.13% |
| L4/N12934+ | ⁺proper nouns or brands indicated by speci... | 0.927 | 2.37% | 1.15% |
| L30/N11430− | ⁻N.A. | 0.969 | 1.76% | 0.89% |
| L20/N7071− | ⁻Arabic phrases, particularly those relate... | 1.000 | 1.76% | 0.00% |
| L30/N13513+ | ⁺activation occurs primarily with tokens i... | 0.000 | −1.62% | 0.00% |
| L30/N6707− | ⁻tokens related to "[CYR:O][CYR:EN][CYR:EL... | 0.080 | −1.55% | −0.29% |
| L24/N9183+ | ⁺activation occurs on connectors like {{... | 0.910 | 1.55% | 0.55% |
| L31/N8309− | ⁻miscellaneous Arabic words and phrases | 0.094 | −1.38% | −0.08% |
| L4/N12458− | ⁻text containing a colon ({{:}}) | 0.906 | 1.29% | 0.94% |
| L31/N10275− | ⁻Greek tokens with certain vowel combinati... | 0.000 | −1.28% | 0.02% |
| L30/N10044− | ⁻presence of Arabic tokens that reflect a ... | 1.000 | 1.17% | 0.00% |
| L31/N3923+ | ⁺Arabic text; often involves instructional... | 0.917 | 1.13% | 0.00% |
| L29/N11490− | ⁻conjunctions and phrases indicating compa... | 1.000 | 1.09% | 0.03% |
| L31/N4130+ | ⁺activation following certain Arabic token... | 0.917 | 1.08% | 0.00% |
| **Target: de** | | | | |
| L31/N1000− | ⁻French or German phrases with hints of co... | 1.000 | 4.54% | 0.00% |
| L0/N11311− | ⁻instances of "{{user}}", mainly in co... | 0.948 | 2.21% | 1.19% |
| L30/N7745+ | ⁺usage of "um" or "ich" in various context... | 0.000 | −1.60% | 0.00% |
| L24/N6638− | ⁻Dutch articles, conjunctions, and demonst... | 1.000 | 1.49% | 0.00% |
| L29/N3490− | ⁻activation occurs when a word signifies t... | 0.962 | 1.24% | 0.13% |
| L23/N1423+ | ⁺conjunctions like {{und}} (and) and ;. | 0.993 | 1.23% | 0.25% |
| L29/N1884+ | ⁺activation occurs primarily on punctuatio... | 0.917 | 0.98% | 0.00% |
| L14/N4513− | ⁻responses indicating inability to perform... | 0.997 | 0.96% | 0.10% |
| L4/N12353− | ⁻phrases indicating a response of "yes" or... | 0.993 | 0.90% | 0.22% |
| L0/N1050+ | ⁺occurrences of "{{Ant}}" in various c... | 1.000 | 0.86% | 0.00% |
| L13/N4776− | ⁻the conjunction "e" or similar connecting... | 0.000 | −0.79% | −0.11% |
| L21/N11206+ | ⁺user queries for summaries, rejections, o... | 0.045 | −0.71% | −0.12% |
| L29/N11794− | ⁻tokens indicating object relations or gra... | 0.972 | 0.65% | 0.14% |
| L12/N12356+ | ⁺translations into or out of Arabic and Po... | 0.990 | 0.58% | −0.02% |
| L10/N9553+ | ⁺tokens related to "download," "slot equip... | 0.003 | −0.56% | −0.03% |
| L21/N5154− | ⁻tokens "something" and "distinctly" in va... | 0.915 | 0.54% | 0.01% |

*Table 10.* Feature scores for language task.

| Feature | Description | AUROC | In-class | Out-of-class |
|---|---|---|---|---|
| L15/N1016+ | $^+$The neuron activates on the token delimit... | 0.092 | −0.47% | −0.03% |
| **Target: en** | | | | |
| L12/N8459− | $^-$responses to inquiries about word associa... | 0.000 | −4.63% | −1.78% |
| L8/N10006− | $^-$group {{of}}; the pattern "called" re... | 0.056 | −3.24% | −1.08% |
| L0/N2765− | $^-$tokens corresponding to "Answer" in vario... | 1.000 | 2.59% | 0.07% |
| L29/N12010− | $^-$Names, articles, or references with forma... | 0.969 | 2.42% | 0.38% |
| L15/N11853+ | $^+$response with {{No}} or {{'m}} (I... | 0.951 | 2.15% | 1.30% |
| L15/N1816− | $^-$tokens that follow a specific format or c... | 1.000 | 1.83% | 0.05% |
| L29/N3490+ | $^+$tokens like "cho," "[CJK]," and "[CJK]" i... | 0.076 | −1.81% | −0.29% |
| L31/N13336− | $^-$references to specific numerical IDs, mon... | 1.000 | 1.81% | 0.18% |
| L12/N14160+ | $^+$usage of ampersand {{&}}, commas bef... | 0.000 | −1.79% | −0.22% |
| L21/N5779− | $^-$presence of "a" or "those" before tokens ... | 0.951 | 1.65% | 1.33% |
| L25/N11584+ | $^+$activating words that indicate a type or ... | 0.000 | −1.62% | −0.09% |
| L13/N9823+ | $^+$the word "that" in definitions and explan... | 1.000 | 1.57% | 0.28% |
| L15/N1101− | $^-$presence of specific tokens like {{*}}.. | 0.000 | −1.42% | 0.00% |
| L12/N5697+ | $^+$occurrences of the token "to" when preced... | 1.000 | 1.38% | 0.01% |
| L21/N4722+ | $^+$activating tokens related to comparative ... | 0.000 | −1.33% | −0.11% |
| L13/N9720+ | $^+$words that imply a certain quality or sta... | 0.000 | −1.32% | 0.00% |
| L19/N464− | $^-$phrases indicating actions or characteris... | 1.000 | 1.25% | 0.88% |
| L30/N936− | $^-$specific numerical patterns (e.g., EAN or... | 1.000 | 1.23% | 0.00% |
| L8/N1789+ | $^+$space and formatting characters (e.g., "[... | 0.087 | −1.11% | −0.79% |
| L12/N10867+ | $^+$tokens indicating ethical constraints and... | 0.000 | −1.08% | −0.09% |
| **Target: es** | | | | |
| L31/N9968+ | $^+$extraction of specific information or req... | 0.944 | 1.54% | 0.16% |
| L31/N11229+ | $^+$conjunctions (e, e para) in academic or i... | 0.965 | 1.45% | 0.15% |
| L28/N5596+ | $^+$presence of defined concepts or terms rel... | 0.927 | 1.35% | 0.32% |
| L30/N8299− | $^-$activation occurs on conjunctions and pre... | 0.917 | 1.34% | 0.00% |
| L29/N7941+ | $^+$chemistry terms with varying parenthetica... | 0.969 | 1.07% | 0.23% |
| L12/N7597− | $^-$tokens indicating mathematical or coding ... | 1.000 | 0.96% | 0.09% |
| L10/N1648− | $^-$the use of the token {{.}} at the end... | 1.000 | 0.92% | 0.04% |
| L10/N6962− | $^-$highlighting of specific affirmative or c... | 0.045 | −0.89% | −0.05% |
| L0/N4686+ | $^+$activation occurs on the token format '{... | 0.983 | 0.87% | 0.09% |
| L10/N10534− | $^-$tokens with specific accented characters ... | 0.000 | −0.79% | 0.00% |
| L10/N14114− | $^-$references to equivalence or corresponden... | 0.969 | 0.77% | 0.10% |
| L31/N11267− | $^-$the activation of the neuron occurs on to... | 0.917 | 0.72% | 0.00% |
| L13/N1134− | $^-$mentions of "Anth" or forms of "we/us" (:. | 0.014 | −0.71% | −0.07% |
| L17/N14096− | $^-$presence of specific nouns related to mea... | 0.903 | 0.66% | 0.13% |
| L9/N10339− | $^-$specific terms or tokens that reference a... | 1.000 | 0.65% | 0.00% |
| L10/N191+ | $^+$abrupts or irrelevant jumps from topic to... | 1.000 | 0.59% | 0.00% |
| L1/N7598+ | $^+$phrases "the point {{being}}", "the g... | 1.000 | 0.58% | 0.00% |
| L5/N12754+ | $^+$Activation occurs on affirmations or nega... | 0.917 | 0.47% | 0.00% |
| **Target: fr** | | | | |
| L26/N808− | $^-$French connectors or conjunctions like {... | 1.000 | 3.19% | 0.00% |
| L31/N5283+ | $^+$introduction of a descriptor or explanati... | 1.000 | 2.87% | 0.00% |
| L5/N2898+ | $^+$a tokenized structure starting with an op... | 0.969 | 0.96% | 0.73% |
| L5/N919+ | $^+$words containing repeated letters or spec... | 0.017 | −0.87% | −0.03% |
| L23/N5245− | $^-$French tokens indicating possession or in... | 0.083 | −0.71% | 0.00% |
| L1/N3234− | $^-$usage of the token "{{:}}" to indicat... | 0.083 | 0.70% | 0.91% |
| L0/N5805− | $^-$occurrences of {{=}} or {{:}} ind... | 1.000 | 0.64% | 0.00% |
| L0/N10902− | $^-$token "{{¡—start_header_id—¿}}" in ... | 1.000 | 0.62% | −0.04% |
| L9/N7246+ | $^+$the token "you" preceded by a context imp... | 0.000 | −0.61% | −0.01% |
| L21/N8322+ | $^+$activating token "env", "que", "sous", an... | 0.083 | −0.53% | 0.00% |
| L31/N8152− | $^-$presence of specific terms related to tec... | 0.917 | 0.50% | 0.00% |
| **Target: hi** | | | | |
| L31/N742− | $^-$hindi tokens involving conjunctions and s... | 1.000 | 24.13% | 0.00% |
| L31/N8104+ | $^+$Hindi phrases and prompts for writing poe... | 0.000 | −3.52% | 0.00% |

*Table 10.* Feature scores for language task.

| Feature | Description | AUROC | In-class | Out-of-class |
|---|---|---|---|---|
| L30/N3382+ | +common contextual words or phrases before... | 0.094 | −3.34% | −0.98% |
| L29/N12010− | ¬Names, articles, or references with forma... | 0.000 | −3.00% | 1.06% |
| L31/N11558− | ¬tokens that initiate a new line or stanza... | 1.000 | 2.60% | 0.00% |
| L8/N11999+ | +names with specific endings like -berg, -... | 1.000 | 1.68% | 0.00% |
| L3/N9171+ | +the letter "A" as an abbreviation for "an... | 1.000 | 1.66% | 0.20% |
| L27/N8140− | ¬words like "meeste", "tienen", "die" that... | 0.045 | −1.64% | −0.02% |
| L5/N8064+ | +tokens related to "sales" and "Medicare" ... | 0.000 | −1.63% | 0.00% |
| L30/N6707− | ¬tokens related to "[CYR:O][CYR:EN][CYR:EL... | 0.045 | −1.62% | −0.28% |
| L21/N5178− | ¬presence of the Hindi tokens for "ham", "... | 1.000 | 1.50% | 0.00% |
| L27/N9595+ | +activation on specific word forms in Sout... | 1.000 | 1.37% | 0.00% |
| L28/N1945− | ¬activation occurs with conjunctions and c... | 0.017 | −1.34% | −0.16% |
| L30/N6764− | ¬presence of a string of characters with f... | 0.083 | −1.28% | 0.00% |
| L30/N2220+ | +presence of characters from the Japanese ... | 0.000 | −1.10% | 0.00% |
| L2/N4313+ | +activations on variations of "anser" show... | 0.924 | 1.08% | 0.50% |
| L29/N10856− | ¬Neuron activates on the presence of suita... | 0.035 | −1.00% | −0.08% |
| L0/N7433+ | +opening curly brace '{{' | 0.938 | 0.95% | 0.21% |
| L15/N4528− | ¬activation on tokens related to user quer... | 1.000 | 0.90% | 0.00% |
| L19/N8664+ | +context featuring social dynamics, person... | 0.924 | −0.88% | −1.54% |
| **Target: it** | | | | |
| L27/N8557− | ¬copulas and auxiliary verbs ([U+00E8], so... | 1.000 | 2.15% | 0.00% |
| L1/N2427+ | +partial URL tokens or incomplete words, o... | 0.000 | −1.73% | −0.34% |
| L31/N11229+ | +conjunctions (e, e para) in academic or i... | 0.910 | 1.23% | 0.18% |
| L23/N6734− | ¬activation occurs on the Italian token "c... | 0.000 | −1.16% | 0.00% |
| L23/N11405− | ¬Month names within a specific context (of... | 0.969 | 1.07% | 0.41% |
| L1/N3234− | ¬usage of the token "{{:}}" to indicat... | 0.924 | 1.06% | 0.87% |
| L1/N198− | ¬system instruction/configuration text | 1.000 | 0.80% | 0.00% |
| L9/N470− | ¬keywords related to a specific identity o... | 0.031 | −0.78% | −0.34% |
| L31/N3561− | ¬verbs like {{usar}}, {{ar}} (from... | 0.917 | 0.60% | 0.00% |
| L5/N13635− | ¬followed by a colon {{:}} and often i... | 0.000 | 0.00% | 0.67% |
| **Target: ru** | | | | |
| L31/N7925− | ¬activation occurs on the pattern of a tok... | 1.000 | 3.45% | 0.00% |
| L31/N8180− | ¬activation occurs on tokens that are non-... | 1.000 | 3.26% | 0.00% |
| L31/N9792+ | +poor grammar, typographical errors, nonse... | 1.000 | 2.73% | 0.00% |
| L30/N10747− | ¬brief summaries or explanations, often in... | 0.000 | −2.37% | 0.00% |
| L30/N13605− | ¬occurrence of Russian tokens related to c... | 1.000 | 2.36% | 0.00% |
| L31/N13501− | ¬conjunctions (e.g. "[CYR:I]", "[CYR:CHE][... | 1.000 | 2.18% | 0.00% |
| L7/N2742− | ¬presence of tokens like "Dur{{ante}}"... | 0.986 | 2.16% | 0.49% |
| L31/N5565− | ¬activation on {{"}} following a comma... | 0.986 | 2.07% | 0.42% |
| L3/N6390− | ¬specific tokens containing words related ... | 0.059 | 2.04% | 3.85% |
| L23/N12573− | ¬emphasizes the definite article or the pr... | 0.021 | −1.98% | −0.68% |
| L31/N10131+ | +occurrence of the Cyrillic letters {{[C... | 0.083 | −1.90% | 0.00% |
| L24/N6445− | ¬tokens "w" and "," followed by grammatica... | 1.000 | 1.74% | 0.00% |
| L29/N13810+ | +presence of specific attributes (e.g., 'g... | 0.010 | −1.70% | −0.55% |
| L30/N3745− | ¬activation occurs after sentences or phra... | 0.000 | −1.65% | 0.00% |
| L21/N4786− | ¬activating phrases contain {{(}}, {:. | 0.976 | 1.62% | 0.24% |
| L24/N7999− | ¬presence of a function indicated by {{(... | 1.000 | 1.52% | 0.00% |
| L31/N311− | ¬presence of punctuation or special charac... | 0.983 | 1.43% | 0.43% |
| L27/N9311− | ¬the presence of words like "[CYR:KA][CYR:... | 1.000 | 1.42% | 0.00% |
| L30/N936+ | +phrases starting with "here are" and spec... | 0.038 | −1.42% | −0.16% |
| L31/N2213− | ¬the presence of additional or unexpected ... | 1.000 | 1.25% | 0.00% |
| **Target: zh** | | | | |
| L31/N8258+ | +the presence of specific Chinese characte... | 1.000 | 3.00% | 0.00% |
| L0/N12829+ | +activation occurs on specific tokens that... | 0.958 | 2.45% | 0.72% |
| L31/N4198− | ¬presence of the token "pij" related to pa... | 1.000 | 2.35% | 0.01% |
| L2/N12721+ | +translation or inquiry involving Chinese ... | 0.917 | 2.22% | 0.00% |
| L31/N8959+ | +presence of specific contextual words (e.... | 0.000 | −2.22% | 0.00% |
| L29/N2007+ | +presence of non-English tokens (e.g., "su... | 0.062 | −2.19% | −0.91% |

*Table 10.* Feature scores for language task.

| Feature | Description | AUROC | In-class | Out-of-class |
|---|---|---|---|---|
| L29/N3490+ | $^+$tokens like "cho," "[CJK]," and "[CJK]" i... | 0.049 | −2.01% | −0.27% |
| L0/N10772+ | $^+$Chinese interrogative phrases starting wi... | 1.000 | 1.68% | 0.00% |
| L18/N9669+ | $^+$questions about Vietnamese language compr... | 0.059 | −1.67% | −0.59% |
| L31/N11489+ | $^+$presence of characters indicative of Chin... | 1.000 | 1.60% | 0.00% |
| L30/N8350− | $^-$repeated phrases with certain keywords li... | 1.000 | 1.59% | 0.00% |
| L31/N8236+ | $^+$activation occurs on tokens that follow s... | 1.000 | 1.38% | 0.00% |
| L1/N11008− | $^-$Chinese characters, particularly {{[CJK... | 1.000 | 1.31% | 0.00% |
| L30/N255+ | $^+$Japanese and Chinese dialogue, activating... | 0.000 | −1.26% | 0.00% |
| L3/N10591− | $^-$mentions of command line syntax or specia... | 0.972 | 1.20% | 0.80% |
| L31/N1443+ | $^+$the presence of specific terms related to... | 0.911 | 1.06% | 0.01% |
| L22/N4433+ | $^+$activation occurs on prepositions, conjun... | 0.097 | −0.98% | −0.24% |
| L0/N5936+ | $^+$repeated token "[CJK][CJK]" | 1.000 | 0.97% | 0.00% |
| L31/N10233+ | $^+$activation occurs on single Chinese chara... | 0.993 | 0.96% | 0.03% |
| L31/N3583− | $^-$the presence of commas in citation format... | 0.997 | 0.94% | 0.10% |

### H.2.2. CONCEPT AND ATTRIBUTE

Concept and attribute neurons are less numerous but again have relevant descriptions that even indicate their multilingual nature, e.g. L2/N1709− which has the description "the word "kalter" or its variants (e.g. "kaltes", "fria") in the context of cold conditions or descriptions of weather", which includes both the German word *kalter* and the Spanish word *fria*. We show all of the neurons exceeding the AUROC threshold for concept and attribute in the two tables below.

*Table 11.* Feature scores for axis task.

| Feature | Description | AUROC | In-class | Out-of-class |
|---|---|---|---|---|
| **Target: size** | | | | |
| L21/N4920− | $^-$activation occurs on the token "from" whe... | 0.054 | −3.15% | −0.77% |
| L25/N14237+ | $^+$the word "pi[U+00F9]" in an Italian conte... | 0.889 | 1.35% | 0.00% |
| L14/N2801− | $^-$tokens "largest," "biggest," "size," "sma... | 1.000 | 1.05% | 0.00% |
| L22/N4433+ | $^+$activation occurs on prepositions, conjun... | 0.185 | −0.71% | −0.29% |
| L22/N929+ | $^+$colloquial use of "be" in contexts discus... | 0.094 | 0.04% | 0.61% |
| L19/N1691+ | $^+$tokens suggesting duality or contrast | 0.889 | 0.00% | −0.53% |
| **Target: speed** | | | | |
| L25/N7423+ | $^+$tokens "m[U+00E1]s" and "pi[U+00F9]" indi... | 0.889 | 5.40% | 0.00% |
| L25/N6793+ | $^+$comparative expressions about speed or pa... | 0.194 | −1.92% | 0.00% |
| L24/N13094+ | $^+$comparative adjectives such as {{m[U+00... | 0.889 | 1.70% | 0.00% |
| L20/N11041+ | $^+$the phrase "at your own" or variations; "... | 0.889 | 1.51% | 0.00% |
| L16/N8263− | $^-$the word {{Fast}} or phrases indicati... | 0.972 | 1.13% | 0.00% |
| L20/N4961+ | $^+$activates on single letters or specific t... | 0.858 | 1.06% | 0.87% |
| L19/N1691+ | $^+$tokens suggesting duality or contrast | 0.182 | −0.66% | −0.20% |
| L21/N351− | $^-$activating tokens include "yang" followin... | 0.808 | 0.63% | 0.13% |
| L19/N1643− | $^-$activation occurs on references to speed ... | 0.167 | −0.45% | 0.00% |
| L18/N1015+ | $^+$context-specific queries regarding notabl... | 0.867 | 0.44% | 0.26% |
| L18/N10753+ | $^+$function words indicating an action or pu... | 0.829 | 0.34% | 0.05% |
| **Target: temperature** | | | | |
| L26/N5654+ | $^+$activation on tokens indicating high temp... | 0.917 | 2.20% | 0.00% |
| L25/N2969− | $^-$tokens related to temperature and warmth/... | 0.139 | −1.60% | 0.00% |
| L19/N5652− | $^-$mentions of heat, temperature, warming, o... | 0.000 | −1.09% | 0.00% |
| L14/N13885− | $^-$temperature unit "{{[U+00B0]C}}" in s... | 1.000 | 1.04% | 0.00% |
| L20/N8802− | $^-$requests for spacing out words or clarifi... | 0.823 | −0.63% | −1.30% |
| L22/N7399− | $^-$tokens "or" and "and" in response to inqu... | 0.805 | 0.49% | 0.11% |
| L20/N7483+ | $^+$the token "{{in}}" in various context... | 0.139 | −0.33% | 0.00% |
| L20/N7973+ | $^+$temperature ranges and conditions | 0.833 | 0.27% | 0.00% |
| L15/N9288+ | $^+$mentions of named entities (people, bands... | 0.810 | 0.25% | 0.02% |
| L17/N4188− | $^-$the word "he" when followed by clauses, i... | 0.804 | −0.14% | −0.41% |
| L21/N4920− | $^-$activation occurs on the token "from" whe... | 0.976 | −0.14% | −2.27% |

*Table 11.* Feature scores for axis task.

| Feature | Description | AUROC | In-class | Out-of-class |
|---|---|---|---|---|
| L22/N2802+ | $^+$highlighting of terms related to potentia... | 0.111 | 0.00% | 0.56% |

*Table 12.* Feature scores for word task.

| Feature | Description | AUROC | In-class | Out-of-class |
|---|---|---|---|---|
| **Target: big** | | | | |
| L27/N3345− | $^-$specific phrases that denote small or med... | 1.000 | 5.00% | 0.00% |
| L21/N4920− | $^-$activation occurs on the token "from" whe... | 0.114 | −3.34% | −1.21% |
| L26/N4089+ | $^+$expressions indicating quantity or qualit... | 0.000 | −2.64% | 0.00% |
| L24/N13961+ | $^+$appearance of words indicating direction ... | 1.000 | 1.56% | 0.00% |
| L14/N2801− | $^-$tokens "largest," "biggest," "size," "sma... | 0.968 | 1.34% | 0.15% |
| L22/N9062+ | $^+$the tokens "about," "of the," "the," "on,... | 0.889 | 0.75% | 0.00% |
| L2/N3728− | $^-$words or phrases indicating "greatness" o... | 0.889 | 0.74% | −0.00% |
| L23/N9337− | $^-$the word "with" when discussing ease or m... | 0.111 | −0.67% | 0.00% |
| L19/N4804− | $^-$conjunctions indicating contrast (e.g., :. | 0.935 | 0.65% | 0.02% |
| L28/N9284+ | $^+$tokens following the phrase "create" or "... | 0.864 | 0.60% | 0.04% |
| L21/N9786+ | $^+$The token "{{[U+FF0C][CJK]}}" activat... | 0.872 | 0.39% | 0.02% |
| L22/N929+ | $^+$colloquial use of "be" in contexts discus... | 0.173 | 0.03% | 0.50% |
| L28/N5029− | $^-$presence of specific keywords/phrases tha... | 0.833 | 0.00% | −0.49% |
| L19/N1691+ | $^+$tokens suggesting duality or contrast | 0.811 | 0.00% | −0.43% |
| **Target: cold** | | | | |
| L24/N10117− | $^-$activating token precedes lists or bullet... | 1.000 | 5.87% | 0.00% |
| L26/N5654+ | $^+$activation on tokens indicating high temp... | 0.988 | 3.82% | 0.12% |
| L25/N9791+ | $^+$words that directly relate to producing o... | 0.944 | 3.56% | 0.00% |
| L25/N2969− | $^-$tokens related to temperature and warmth/... | 0.007 | −2.75% | −0.09% |
| L2/N1709− | $^-$the word "kalter" or its variants (e.g. "... | 0.943 | 2.65% | 0.01% |
| L30/N9108− | $^-$references to keeping warm during cold se... | 0.111 | −2.39% | 0.00% |
| L14/N13885− | $^-$temperature unit "{{[U+00B0]C}}" in s... | 0.948 | 1.17% | 0.18% |
| L19/N5652− | $^-$mentions of heat, temperature, warming, o... | 0.101 | −1.10% | −0.22% |
| L26/N3401− | $^-$occurences of "is", "one" in various cont... | 0.830 | 0.55% | 0.01% |
| L15/N6003− | $^-$words and phrases related to coldness ({... | 0.833 | 0.35% | 0.00% |
| L20/N7973+ | $^+$temperature ranges and conditions | 0.849 | 0.34% | 0.04% |
| L21/N4920− | $^-$activation occurs on the token "from" whe... | 0.838 | −0.23% | −1.83% |
| L22/N2802+ | $^+$highlighting of terms related to potentia... | 0.189 | 0.00% | 0.45% |
| **Target: fast** | | | | |
| L26/N878+ | $^+$the word "[CYR:I]" and "[CYR:TE]" in conj... | 1.000 | 15.65% | 0.00% |
| L20/N11041+ | $^+$the phrase "at your own" or variations; "... | 1.000 | 2.71% | 0.06% |
| L27/N13178− | $^-$occurrence of the word "be" | 0.849 | 1.17% | 0.45% |
| L20/N4961+ | $^+$activates on single letters or specific t... | 0.847 | 1.10% | 0.90% |
| L16/N8263− | $^-$the word {{Fast}} or phrases indicati... | 0.820 | 0.97% | 0.26% |
| L22/N2802+ | $^+$highlighting of terms related to potentia... | 0.806 | 0.89% | 0.27% |
| L19/N1691+ | $^+$tokens suggesting duality or contrast | 0.096 | −0.87% | −0.25% |
| L28/N5029− | $^-$presence of specific keywords/phrases tha... | 0.146 | −0.82% | −0.32% |
| L19/N1643− | $^-$activation occurs on references to speed ... | 0.138 | −0.66% | −0.05% |
| L23/N4239+ | $^+$emphasis on adjectives like "more," "the,... | 0.830 | 0.59% | 0.18% |
| L14/N5039+ | $^+$the token {{fast}} or its variants (:. | 0.943 | 0.59% | 0.01% |
| L18/N1015+ | $^+$context-specific queries regarding notabl... | 0.923 | 0.50% | 0.28% |
| L13/N9722+ | $^+$rough or casual conversational tone; info... | 0.153 | −0.47% | −0.32% |
| L18/N10753+ | $^+$function words indicating an action or pu... | 0.885 | 0.45% | 0.09% |
| L29/N13822− | $^-$references to roles (like "model", "at", ... | 0.185 | −0.41% | −0.08% |
| L18/N7153+ | $^+$occurrences of the token "Quick" allowing... | 0.944 | 0.40% | 0.00% |
| L22/N7122− | $^-$tokens "take" or "taking" appearing in co... | 0.111 | −0.36% | 0.00% |
| L2/N6829+ | $^+$occurrences of "miles per {{hour}}" a... | 0.804 | 0.35% | 0.03% |
| L27/N13857− | $^-$occurrence of delimiters "{{}}" indic... | 0.190 | 0.12% | 0.40% |
| **Target: hot** | | | | |

*Table 12.* Feature scores for word task.

| Feature | Description | AUROC | In-class | Out-of-class |
|---------|-------------|-------|----------|--------------|
| L22/N1649− | ⁻activation occurs with the word "is" in c... | 1.000 | 5.20% | 0.00% |
| L26/N10210+ | ⁺references to coldness or emotional detac... | 0.056 | −1.68% | 0.00% |
| L16/N1694+ | ⁺the word "hot" or its variants in diverse... | 1.000 | 1.66% | 0.00% |
| L27/N6103+ | ⁺activation occurs on the token "to" when ... | 0.889 | 1.65% | 0.00% |
| L2/N7012− | ⁻presence of "calent", "calore", "insel", ... | 1.000 | 1.38% | 0.00% |
| L19/N5652− | ⁻mentions of heat, temperature, warming, o... | 0.099 | −1.08% | −0.22% |
| L14/N13885− | ⁻temperature unit "{{[U+00B0]C}}" in s... | 0.852 | 0.91% | 0.23% |
| L21/N6962+ | ⁺Tokens indicating states of comfort, seek... | 1.000 | 0.86% | 0.00% |
| L19/N2619+ | ⁺references to sensations related to heat,... | 0.804 | 0.61% | 0.06% |
| L0/N12140+ | ⁺the word "Hoh{{hot}}" when mentioned ... | 0.833 | 0.60% | 0.00% |
| L21/N11022− | ⁻"in" or "at" when used in contexts relate... | 0.111 | −0.58% | 0.00% |
| L20/N7483+ | ⁺the token "{{in}}" in various context... | 0.165 | −0.39% | −0.05% |
| L27/N6438+ | ⁺activation occurs on the token "and" and ... | 0.833 | 0.39% | 0.00% |
| L19/N5170+ | ⁺negation or conditions such as unsure or ... | 0.131 | −0.30% | −0.02% |
| L24/N7819+ | ⁺words related to heat or melting, especia... | 0.177 | −0.29% | −0.01% |
| L26/N10433− | ⁻the token "been," the word "very," and th... | 0.833 | 0.25% | 0.00% |
| L24/N2804− | ⁻the word "like" or "for" following conver... | 0.199 | 0.08% | 0.35% |
| L21/N4920− | ⁻activation occurs on the token "from" whe... | 0.923 | −0.04% | −1.86% |
| L22/N2802+ | ⁺highlighting of terms related to potentia... | 0.189 | 0.00% | 0.45% |
| **Target: slow** | | | | |
| L25/N7423+ | ⁺tokens "m[U+00E1]s" and "pi[U+00F9]" indi... | 1.000 | 10.55% | 0.05% |
| L25/N6793+ | ⁺comparative expressions about speed or pa... | 0.000 | −3.67% | −0.03% |
| L24/N13094+ | ⁺comparative adjectives such as {{m[U+00... | 0.973 | 2.55% | 0.17% |
| L3/N738+ | ⁺tokens indicating a slow action or pace (... | 1.000 | 1.65% | 0.00% |
| L16/N8263− | ⁻the word {{Fast}} or phrases indicati... | 0.936 | 1.29% | 0.19% |
| L14/N9193+ | ⁺the token "slow" and its variations (e.g.... | 0.944 | 1.07% | 0.00% |
| L29/N8749+ | ⁺words that denote the degree of compariso... | 0.833 | 0.90% | 0.00% |
| L26/N4919+ | ⁺words indicating speed or quickness (e.g.... | 0.056 | −0.75% | 0.00% |
| L23/N5561+ | ⁺phrases indicating a lack of hurry or ind... | 0.944 | 0.63% | 0.00% |
| L24/N4630− | ⁻presence of a token indicating immediacy ... | 0.833 | 0.62% | 0.00% |
| L21/N8063− | ⁻the word "the" when discussing speed or f... | 0.889 | 0.39% | 0.00% |
| L22/N3698+ | ⁺comparative terms and evaluative triggers... | 0.151 | −0.36% | −0.04% |
| **Target: small** | | | | |
| L21/N4920− | ⁻activation occurs on the token "from" whe... | 0.173 | −2.96% | −1.28% |
| L26/N2892− | ⁻tokens indicating a specific or superlati... | 1.000 | 2.84% | 0.01% |
| L29/N3261+ | ⁺occurrence of the word "all" or words ind... | 0.000 | −2.16% | 0.00% |
| L25/N14237+ | ⁺the word "pi[U+00F9]" in an Italian conte... | 0.832 | 1.55% | 0.23% |
| L28/N2402− | ⁻queries about size comparisons and enlarg... | 0.884 | 1.28% | 0.02% |
| L27/N13928− | ⁻Token "large"/"giant" following compariso... | 0.000 | −1.24% | 0.00% |
| L22/N4433+ | ⁺activation occurs on prepositions, conjun... | 0.185 | −0.82% | −0.35% |
| L14/N2801− | ⁻tokens "largest," "biggest," "size," "sma... | 0.832 | 0.75% | 0.27% |
| L14/N8678− | ⁻references to smallness or reduction (e.g... | 0.889 | 0.56% | 0.00% |
| L17/N4188− | ⁻the word "he" when followed by clauses, i... | 0.195 | −0.56% | −0.28% |
| L18/N12178− | ⁻the token "a" and "or" trigger activation... | 0.886 | 0.50% | 0.01% |
| L17/N6931− | ⁻the token "Small" when it is the only wor... | 0.864 | 0.43% | 0.03% |
| L22/N9062− | ⁻activation on specific tokens (e.g., "as"... | 0.889 | 0.36% | 0.00% |
| L24/N9648+ | ⁺occurrences of non-English language phras... | 0.891 | −0.07% | −0.49% |
| L23/N7598− | ⁻a specific token (or token sequence) that... | 0.827 | −0.04% | −0.59% |
| L22/N929+ | ⁺colloquial use of "be" in contexts discus... | 0.178 | 0.04% | 0.49% |
| L19/N1691+ | ⁺tokens suggesting duality or contrast | 0.811 | 0.00% | −0.43% |

## H.3. User modelling

Finally, we investigate a new task: user modelling. The user model is the set of inferences a language model makes about the user during their interaction; since these inferences may encoded protected demographic attributes which the user may want to know about or adjust, prior work has argued that surfacing the user model to the end user is an ethical need (Viégas

& Wattenberg, 2023; Chen et al., 2024). We thus investigate whether neuron-level circuit tracing can find features which encode user model inferences.

**Dataset.** We construct a relatively contrived dataset for this case study: since we must trace our circuit back from actual output logits, we design our dataset such that it forces the model to explicitly state its belief about the user's demographics by prefilling the assistant response. Specifically, our dataset contains examples where the user shares a fact about themselves, and then asks the model to construct the biographical infobox that would be present in a hypothetical Wikipedia article about them. Since these infoboxes contain structured fields for attributes like gender, country of origin, occupation, and religion, we can easily make the model generate these attributes as output tokens.

An example from the gender subset (which covers male, female, and non-binary users) is shown below:

---

**User modeling from preferences**

**User**   I hit the gym regularly, where I focus on lifting heavy weights and achieving my personal bests. Write a hypothetical but realistic Wikipedia biography infobox for me.

**Assistant**   {{Infobox person\n| gender =_Male

---

For instance, given this synthetic input from a user describing their preference for gym activities and lifting heavy weights, the Assistant's task is to predict the user's gender in a Wikipedia infobox format. We then use the predicted token (_Male) to trace the circuit that contributes to this user modeling behavior.

**Analysis.** We show the neurons which achieve AUROCs greater than 0.8 or less than 0.2, and which have in-class average attribution scores greater than 0.025 in the table below. Note that these neurons have relevant descriptions in each of the three gender classes.

In general, we note that the model's inferences are not stored in a fixed token position, which makes it difficult to access its belief state. We leave further investigation of the user modelling circuit to future work.

*Table 13.* Feature scores for gender task.

| Feature | Description | AUROC | In-class | Out-of-class |
|---|---|---|---|---|
| **Target: female** | | | | |
| L12/N13860+ | [+]The neuron activates on tokens that are i... | 0.820 | 6.85% | 5.79% |
| L25/N5082− | [−]tokens representing a gender indicator li... | 0.902 | 5.00% | 2.27% |
| **Target: male** | | | | |
| L23/N1638+ | [+]the word "of" indicating a proportion or ... | 0.155 | −5.46% | −2.42% |
| L24/N4684+ | [+]presence of specific gender markers or id... | 1.000 | 5.18% | 0.98% |
| L0/N6791+ | [+]tokens related to "gender" as "{{gender... | 0.833 | 4.21% | 3.26% |
| L30/N10339− | [−]activating tokens "the" and "a" in contex... | 0.090 | −4.05% | −1.30% |
| L27/N10585+ | [+]activation on gender markers {{:}} an... | 0.876 | 3.61% | 0.02% |
| L14/N11231+ | [+]terms referring to {{gender}} or {{... | 0.823 | 3.21% | 2.47% |
| L22/N252− | [−]activating tokens refer to descriptions o... | 0.882 | 2.80% | 1.51% |
| **Target: non-binary** | | | | |
| L18/N7765+ | [+]references to gender identity and pronoun... | 0.999 | 10.00% | 1.81% |
| L9/N4255+ | [+]N.A. | 0.012 | 9.72% | 13.50% |
| L6/N5866− | [−]N.A. | 0.015 | 9.61% | 13.28% |
| L11/N11321+ | [+]N.A. | 0.047 | 9.41% | 12.41% |
| L7/N6673− | [−]N.A. | 0.005 | 7.74% | 10.31% |
| L5/N7012− | [−]N.A. | 0.018 | 7.68% | 11.40% |
| L23/N7782− | [−]the keyword "for", "as", [U+201C]or a[U+2... | 1.000 | 6.68% | 1.07% |
| L10/N11570− | [−]N.A. | 0.014 | 6.24% | 10.73% |
| L12/N13860+ | [+]The neuron activates on tokens that are i... | 0.190 | 5.35% | 6.48% |
| L31/N2293+ | [+]tokens indicating work, identity, or loca... | 0.942 | 4.37% | 0.84% |
| L30/N4121+ | [+]segments indicating addition or presence,... | 0.000 | −4.34% | −0.18% |
| L3/N6390− | [−]specific tokens containing words related ... | 0.086 | 3.59% | 4.66% |
| L29/N7455+ | [+]occurrence of the token "ou" in French qu... | 0.960 | 3.37% | 0.53% |

*Table 13.* Feature scores for gender task.

| Feature | Description | AUROC | In-class | Out-of-class |
|---------|-------------|-------|----------|--------------|
| L2/N4786- | [-]N.A. | 0.140 | 2.94% | 3.68% |
| L20/N2837+ | [+]references to LGBTQ+ identities and exper... | 0.999 | 2.69% | 0.11% |
| L0/N6791+ | [+]tokens related to "gender" as "{{gender... | 0.010 | 2.61% | 4.11% |

