# OpenReview forum: "Language Model Circuits Are Sparse in the Neuron Basis"
_ICML.cc/2026/Conference — ICML 2026 spotlight_

### Official Review · Reviewer_pgYG · 2026-03-09

**Soundness:** 4
**Presentation:** 2
**Significance:** 4
**Originality:** 4
**Overall Recommendation:** 5
**Confidence:** 4

**Summary:**

This work provides an insightful and interesting finding that some of the neurons' information inside LLMs does not need further decomposition via techniques like SAE to exhibit interpretability. Through this finding, one can establish a distinct circuit tracing pipeline by only looking at the neuron bias with certain gradient information. More surprisingly, circuits built by neuron bases are sufficiently sparse and faithful compared to circuits from SAE features.

**Compliance With Llm Reviewing Policy:**

Affirmed.

**Key Questions For Authors:**

I am supporting the paper acceptance at this stage, and my questions can be found in *Strengths And Weaknesses - Presentation - Weakness*. I am more than willing to raise my assessment if the author team actively resolves my issues.

**Limitations:**

yes

**Strengths And Weaknesses:**

*I would like to express my appreciation and excitement for reviewing this paper first, and I am prone to support acceptance, as I do find lots of insightful points when reviewing this paper.*

## **Soundness**
### **Strengths**
1. Reasonable metric definition and usage: This paper mainly questions whether neuron bases can act as effective components for the circuit tracing task, so they adopt the commonly proven metrics: faithfulness and completeness, with concise yet intuitive definitions.

2. Experiment design for showing the effectiveness: The authors use the SVA benchmark, including paired and unpaired settings, to show that neuron bases can be effective under different task settings. On the other hand, the authors also provide studies for attribute-score methods to further support the main claim. The case study also offers a more generalized version of deploying the proposed finding.

### **Weakness**
I do **NOT** see a significant weakness in terms of soundness dimension, as the paper mainly follows experimental settings from other circuit tracing papers, and all the experiments shown in the paper do contribute to the theme collectively.

## **Presentation**
### **Strength**
Unlike the almost excellent soundness, the current presentation reads more like a technical report and may require further revision. In other words, the paper provides enough details for methods and experiments, but I prefer to have more straightforward analyses and discussions.

### **Weakness**
1. Figures are somehow hard to interpret: Most of the figures mess up with different implementations or feature uses, making it especially difficult to understand the figure without sufficient explanation in captions and context. Moreover, the context of the paper also lacks enough discussion of those figures. For example, why does the completeness metric decrease while the faithfulness metric remains the same? MLP activation can reach faithfulness greater than 1, indicating the existence of harmful noise inside other MLP activation, but why is such noisy indication not revealed in the completeness metric (i.e., a much sharper decrease)? The meaning of Figure 7 and the discussion for Figure 7 are very unclear, which makes the meaning of the case study vague.

2. Paper organization needs some revision: In the main context, the authors only report the experiments held on Llama 3.1 8B, but the paper does provide other LLMs like Gemma 2 2B and 9B. I would suggest moving the experiments and discussion of Gemma models to the main context. Also, section F in the appendix is a crucial discussion and worth presenting in the main context.

## **Significance**
### **Strength**
I believe that this paper has a high impact on the community of mechanism interpretability. In the paper, the authors have shown that some neurons carry interpretable meaning and questioned the motivation for applying external tools like SAE, which require extra training and calibration. By taking a deeper, more cautious look at the model's internal attributes, we might discover broader applications of interpretability.

## **Originality**
This work provides new insights, deepens understanding, and highlights important properties of existing methods.

---

> ### Author Rebuttal · Authors · 2026-03-31
>
> Thank you for your positive review! We’re glad to find you appreciate the work! We’ll do our best to address your points.
>
> Presentation weaknesses:
> 1. Another reviewer mentioned this as well, we’ll add large titles to the figures to help make this clearer; in our desire to be thorough with ablations we missed how dense the figures get. In general, for faithfulness vs. completeness we believe they behave differently due to the stacking nonlinearities of the model; ablating a set vs. ablating its complement does not simply result in opposite behaviours. You can consider a toy case of a model $x \times y$ with baseline values $x=0, y=0$. Ablating by setting $x=1$ or $y=1$ alone is insufficient to change the output. However, if your baseline is $x=1, y=1$, then ablating either to $x=0$ etc. is enough to change the output. Imagine having a deep neural network with many such nonlinearities and it should be clear that ablation on a set vs. its complement need not result in complementary outputs.
>     1. Re: Figure 7, we mainly want to show that changing the relative activation of a single neuron is enough to change outputs in an interpretable way across the dataset. Each line is the effect of steering that neuron shown on a single input example from the capital city dataset. Our presentation may be a bit rushed so we will try to clarify the exact procedure (Equation 20 is the main operation being done to the neuron) and label the figure better, as other reviewers have requested the same.
> 2. We moved Gemma to the appendix since the results match Llama and we were tight on space; given camera ready space increase we can consider moving them to the main text! Same for Appendix F; we felt it shows interesting new information about MLP activations vs. other bases but it doesn’t change the conclusions in the main text and we had limited space.
>
> Thank you again!

---

> > ### Author Rebuttal · Reviewer_pgYG · 2026-03-31
> >
> > For my initial perspective, I don't see a major weakness for this submission; on the contrary, its strengths are more obvious. Moreover, the authors also resolve concerns raised by other reviewers actively and effectively (in my opinion). Thus, I would be glad to increase my rating to **5**.

---

### Official Review · Reviewer_Fz2t · 2026-03-12

**Soundness:** 3
**Presentation:** 4
**Significance:** 4
**Originality:** 3
**Overall Recommendation:** 5
**Confidence:** 3

**Summary:**

Authors in this paper claim that neurons are sparse because MLP activations are a privileged basis, therefore can be used in circuit tracing. Existing LLM interpretability research uses Sparse Auto Encoders (SAEs) to learn feature bases from MLP outputs, which can be affected by reconstruction errors and incur training cost for training SAEs. Whereas in the proposed MLP activations don’t require training and reconstruction, as they use feature basis directly from the MLP activations. The second claim is to use more efficient ReLP instead of Integrated Circuits, which closes the gap between MLP SAEs and  MLP activations. They evaluated their claims and approaches on Llama 3.1 8B base model with subject-verb agreement (SVA) benchmarks on 4 templatic datastes with both paired and unparied data. Circuits identified with MLP activations are sparser than

**Compliance With Llm Reviewing Policy:**

Affirmed.

**Final Justification:**

Thanks for the authors' response to my questions. They addressed my questions. And hence raising my score.

**Key Questions For Authors:**

1. A comment on omitting SAEs trained on MLP activations in figures 1-5. is it a cost (multiple passes and large MLP activation size) vs interpretability?
2. In Figure 1, SAEs on attention outputs show negative values for both faithfulness and completeness at small circuit sizes.  a comment on this phenomenon
3. In Figure 2 SAE MLP outputs and Neurons MLP activations achieve faitfulness >1 and completeness close to zero at the same sparsity. A comment on this phenomenon and clarifying the line number 212 in section 5.1

**Limitations:**

Yes

**Strengths And Weaknesses:**

Strengths:

The claim that neuron bases are sparse because it is privileged (next to a gated activation) and hence can be used as a feature for interpretability is novel and very well presented.
The background section is through.
Using 7 representations for learning basis is very thorough and shows circuits that are traced with MLP activations are orders of magnitude sparser.
Case study section 6 with multi-hop reasoning is very well presented which explains the neuron basis for explainability in LLMs.
In section 6, reproducing the findings of Cross-Layer transcoders (CLTs) is very well done. And achieving similar explainability with efficient ReLP and sparser circuit compared to the multiple backpasses IG approach is very good.
Categories in Figure 6 matching with CLT approach not only prove the soundness of the approach but also give striking insight into LLM explainability, as these are from different models.

Weaknesses:

In Figure 1 SAE MLP outputs and Neurons MLP activations achieve faitfulness >1 and completeness close to zero at same spasity which is  not consistent with the claim in text on line number 212 in section 5.1.

In Figure 1, MLP activations achieve faithfulness values up to 1.5. A comment on this phenomenon and whether this is occurring on a specific templatic dataset?

In the results in figures 1-5, one representation of SAEs trained on MLP activations is missing, as it seems a natural comparison to complete the picture.

---

> ### Author Rebuttal · Authors · 2026-03-31
>
> Thank you for your thoughtful review! You raised some good points; we address weakness and questions point-by-point below.
>
> Weaknesses:
> 1. Sorry, for 5.1 text we meant by our claim that using *neuron* MLP activations instead of *neuron* MLP outputs yields 100x smaller circuits, and that this makes neurons more competitive with SAEs. Our claim was not about SAE vs. neuron for the 100x point. We will clarify this in the text.
> 2. We are also puzzled by this phenomenon and do not have a strong explanation for it at this point. Reviewer pgYG suggests this is “indicating the existence of harmful noise inside other MLP activation” which is plausible; it could be that many weakly activating MLP neurons are slightly adjusting predictions, so our attribution metric will first find all helpful noisy neurons (which pushes faithfulness up artificially) and only after that include the harmful noisy neurons for our metric. But this just needs more investigation; it could be related to the smooth form of the SiLU nonlinearity, where negative inputs close to zero are not fully turned off like with ReLU? Perhaps encouraging noise.
> 3. We agree this would be a super important comparison! However it seems that no one has trained MLP activation SAEs for LlamaScope and GemmaScope and we did not have the computational resources to do this ourselves in a comparable manner; we do note in our discussion that there was “earlier use of the MLP activations in interpretability research (Bricken et al., 2023; Gurnee et al., 2023)” but just not recently. Also, MLP activation SAEs would have greater dimensionality than MLP activation neurons, so that should be taken into account in a comparison (the MLP output SAEs are already 2x wider in our experiments). It would definitely be a useful comparison though.
>
> Questions:
> 1. Yes as discussed in #3 above we didn’t train any of our own SAEs etc.
> 2. We are also not sure why this happens but plausibly this is a problem with Integrated Gradients via softmax nonlinearity, which has been empirically observed in prior works (e.g. [Edin et al., 2025](https://arxiv.org/abs/2505.17630)) and possibly happens due to sharp changes in softmax outputs when interpolating for IG. Perhaps the attribution sign by IG is flipped due to this and very harmful attention neurons get included. Attention output representations also have different characteristics than MLP output representations generally, see [Wang et al., 2025](https://arxiv.org/abs/2508.16929v1).
> 3. Discussed above in #1.

---

> > ### Author Rebuttal · Reviewer_Fz2t · 2026-04-04
> >
> > Thank you for the detailed responses and for clarifying the key claims and limitations of the comparable baselines.
> >
> > Overall, I find the paper to make a strong contribution, particularly the case study on multi-hop reasoning, which is well-motivated and insightful. Based on these clarifications, I am inclined to increase my score to Accept.

---

### Official Review · Reviewer_VyrV · 2026-03-13

**Soundness:** 2
**Presentation:** 2
**Significance:** 2
**Originality:** 2
**Overall Recommendation:** 4
**Confidence:** 2

**Summary:**

This paper challenges the common understanding that sparse autoencoder features are actually sparser than the original MLP activations. Through experiments, the authors show that it is possible to obtain the same performance (in terms of faithfulness) as sparse autoencoders by using MLP activations. Furthermore, the authors propose to use RelP instead of the more commonly used Integrated Gradients for circuit discovery, since the latter is more expensive to compute and yields comparable-to-worse results.

**Compliance With Llm Reviewing Policy:**

Affirmed.

**Final Justification:**

The authors have comprehensively answered my points: they motivated the why of certain decisions, and explined points that were not clear after reading the paper.


I had therefore raised my score from 3 to 4.

**Key Questions For Authors:**

**Q1:** In Eq. 1, should $V$ be replaced by $\bar{V}$? Is this a typo?

**Q2:** Can you please clarify what you mean by *feature importance* at the beginning of Sec 5.2?

**Q3:** Are you actually sue the model *must* follow the reasoning chain "Dallas->Texas->Austin"? Why is it not possible that the model learns to automatically map Dallas to Austin whenever the question is about retrieving state capitals? If this is correct, how would it affect the provided evaluation?

**Limitations:**

Yes

**Strengths And Weaknesses:**

**Strengths**

- The paper is mostly clear

- The paper is well structured

- The paper challenges a common assumption in Mech. Inter., and the question it poses is interesting


**Weaknesses**

**W1**: The authors should better motivate some design choices, like:

- why $m$ is chosen to be the logit difference
- why IG is computed by interpolating between $x$ and $x'$
- why mean ablation is applied instead of setting the values to the values they would have taken if a counterfactual input were provided (as discussed in Mueller et al 2025)
- why it is needed to replace the nonlinearities with linear approximations

**W2**: The results in Section 5 should be better contextualized wrt the Figure. For example, by indicating where in the figure certain claims can be located, like the 100x reduction in circuit size.

**W3**: The use of the counterfactual input in the metric computation is unclear. Did you perhaps mean to replace the *x* in the second term of Eq. 5 with *x'*? Similarly, Eq. 11 does not seem to use *x'* at all.

**W4**: The results of Section 6.1 seem a bit handcrafted. In fact, the experiments are provided only for one single case (Texas) and include several manual and subjective choices, like removing neurons from the circuit (Appx. G.1), and identifying the 23 neurons with the best descriptions in line 377.

**W5**: The paper would benefit from including more details about how Figure 6 is made. For instance, it is not clear how the distribution over tokens, represented with a grey box next to each neuron cluster, is actually computed. This might seem trivial for a familiar reader, but it makes it hard to have a clear picture of the experimental section for others.

---

> ### Author Rebuttal · Authors · 2026-03-31
>
> Thank you for your engagement with our work and thorough review! We want to discuss some of the weaknesses you have raised (and try to better clarify them in the text in order to improve the paper) and will also answer your questions. For weaknesses:
>
> 1. These design decisions are largely part of the benchmark or prior work; still, we should justify them since alternatives may exist:
>     1. Using a locally linear approximation (Taylor expansion) of the grad for nonlinearities is done in RelP (Jafari et al., 2025) but has a much longer history with classic interpretability methods like LIME ([Ribeiro et al., 2016](https://arxiv.org/abs/1602.04938)). Empirically we find that RelP helps faithfulness/completeness compared to IG (and IG doesn’t do linear replacement) as our experiments in section 5.2 show. Beyond Taylor expansions being computationally cheaper, we don’t have an axiomatic argument for why RelP is desirable to use; it seems likely that gradients through nonlinearities behave too non-smoothly when doing IG-style interpolation; compare work like [Edin et al., 2025](https://arxiv.org/abs/2505.17630) which observes this.
>     2. The metric being the logit difference, and the use of mean ablation in the SVA benchmark are decisions made in Marks et al. (2025). Metrics like KL or JS divergence seem plausibly just as reasonable but we stuck to the original benchmark parameters for comparability with prior results. We include experiments with zero ablation in Appendix D with similar relative results as mean ablation.
>     3. Integrated Gradients is standardly defined as interpolating between counterfactual pairs $x$ and $x’$ in the original paper [(Sundararajan et al., 2017)](https://proceedings.mlr.press/v70/sundararajan17a.html) with the intuition being that IG distributes credit for a metric difference between $m(x)$ and $m(x’)$ to components via a continuous extension of the Shapley value; it is unclear what alternative one would use since this is part of the core algorithm. We do discuss IG more from a mathematical point of view in Appendix A but didn’t include it in the main text since it is tangential to our result.
> 2. This is a great point; we will add large titles to the figures in the camera-ready and indicate the delta improvements clearly.
> 3. For the equations:
>     1. Equation 5 is correct as written per Marks et al. (2025); the counterfactual only provides the target logits to compare. The idea is that if you ablate the model, the delta between logits for $y$ and $y’$ will become arbitrary and no longer include the input-specific information. Restoring $C$ should help make the delta return to its original value.
>     2. Equation 11 does implicitly include $x’$ since it references $\delta v$ which is defined in Equation 9, but this is definitely unclear since (9) is on an earlier page so we’ll fix the notation to write out the delta.
> 4. We agree that these choices are handcrafted. Ameisen et al. (2025) whose study we wanted to replicate did a similar level of manual inspecting of the circuit for SAE features, including manually grouping features and producing descriptions manually (whereas we use automated feature descriptions from Choi et al. 2024). Ideally this whole process should be automatable and scalable to arbitrary datasets; we have ongoing work on this, but we felt that including it here would detract from our clean and clear result that MLP neurons provide just as interpretable circuits as SAE features, which we show both *quantitatively* (per SVA benchmark results) and *qualitatively* (per the capitals case study). The quantitative results are the main evidence for our claims that the MLP activation neurons are already a strong feature basis for circuit tracing and the qualitative analysis of capitals is only meant to illustrate the downstream applications of this result.
> 5. The token probabilities are steering results derived via the operation in Equation 20, but you are fair to say that this is not clearly indicated in the figure; we will make the connection more explicit by adding some explanatory text under the “steering results” label + arrow.
>
> As for questions:
> 1. Yes that should be $\overline{V}$, thank you for catching this! Definitely a typo.
> 2. We mean Equation 6; we’ll update the text to refer to that and instead say *attribution score* in order to be clearer.
> 3. This is a fair point, the model could certainly just memorise city->state->capital mappings entirely without doing any internal reasoning. If this did happen, we would fail to find the result of figure 7 since there would be no intermediate components engaging in state->capital city mapping since the state would not have to be computed. Therefore, our steering results in figure 7 are evidence that the multi-hop reasoning is actually happening internally. We can mention this in the text clearly.

---

> > ### Author Rebuttal · Reviewer_VyrV · 2026-04-01
> >
> > The authors answered all my questions at an adequate level of detail. Since I do not have major concerns regarding the core contribution of the paper, I'll increase my score from 3 to 4.

---

### Official Review · Reviewer_VC5S · 2026-03-16

**Soundness:** 3
**Presentation:** 3
**Significance:** 3
**Originality:** 3
**Overall Recommendation:** 5
**Confidence:** 3

**Summary:**

This paper investigates different approaches to finding circuits that are responsible for subject-verb agreement (SVA). The main claim of the paper is that the MLP neurons (before the down projection) form sparse feature basis. To show this, the authors conduct comprehensive experiments using different approaches (e.g., neurons vs. SAE, activations at different positions, IG vs RelP, paired and unpaired settings). The do an in-depth analysis on the multi-hop reasoning task.

**Compliance With Llm Reviewing Policy:**

Affirmed.

**Final Justification:**

The rebuttal addressed my main concerns.

**Key Questions For Authors:**

- The main claim is only based on the SVA dataset and a case study on a multi-hop reasoning task. How generalizable is the claim and how robust will the main claim be?
- SAEs are motivated by the issue of superposition. Based on your findings, what can you say about this superposition issue? Is the issue not present not as severe as people expected to be? Or is it the specific dataset (i.e., SVA) that makes the superposition issue less prominent?

Minor comments:
- L051 (right): parse -> sparse?
- L155 (right): Eq. 8 has one extra closing parenthesis in the denominator.
- L259 (right): The paragraph could be merged with previous paragraphs. Note that Figure 3 was already for unpaired setting and this has to be mentioned.
- L328 (right): there is a missing reference "??".
- L378 (left): Stating k=5 may be redundant as "top 5" is already mentioned in L372
- L403 (left): It is unclear to read the alpha from the figure (only alpha=0 and -4 are shown?)

**Limitations:**

No. The main claims in the paper, as with many other ML papers, are subject to the datasets that are used to support the claims. In this paper only the SVA dataset is used. Without corroborating the main claims with experiments on other datasets, their generalizability is not necessarily granted.

**Strengths And Weaknesses:**

The main finding of the paper, that MLP neurons form sparse feature basis, is likely to be useful for future research in mechanistic interpretability. This main finding has to be appreciated with the limitation that the experiments were conducted only on the SVA task with a case study on a multi-hop reasoning task. To show the generalizability of the finding, one would need to include more diverse datasets. The main claim and the methodology are, however, sufficiently interesting and inspiring.

---

> ### Author Rebuttal · Authors · 2026-03-31
>
> Thank you for your questions (and for catching our typos--we will fix those)! To answer them:
>
> 1. We agree that more evaluation benchmarks would be useful additions. Alternatives like the Mechanistic Interpretability Benchmark (which we report scores on in the Appendix) are not designed to evaluate models at the granularity of features (instead, whole MLP blocks and attention heads). Ultimately our field needs a more diverse set of benchmarks for circuit tracing at the feature level. We wanted to make the focus of our paper be the novel result on MLP neurons so we didn’t introduce new evaluations.
>     1. For this rebuttal, we did test running the same comparison (MLP activations neurons w/ RelP vs. SAE MLP outputs w/ IG) on 2 subtasks (`agr_sv_num_pp`, `npi_any_subj`) in CausalGym ([Arora et al., 2024](https://aclanthology.org/2024.acl-long.785/)) as well, and we obtain similar results as for SVA. We will include these in an Appendix in the camera-ready.
> 2. A key feature of the MLP activations which seems to lead to our results is that they are a privileged basis; the preceding SiLU/nonlinearity encourages features to align to the standard basis. This is why we may get more monosemantic neurons in MLPs. In a toy setting, earlier literature on superposition has studied this: [section 7 of Elhage et al. (2022)](https://transformer-circuits.pub/2022/toy_model/index.html#privileged-basis) already notes that superposition in a privileged basis for their toy autoencoder model follows different trends than superposition in non-privileged bases. So this finding is compatible with superposition theoretically. The crucial question is whether MLP activations don’t suffer from superposition; [Gurnee et al., 2023](https://arxiv.org/abs/2305.01610) has some results finding polysemantic MLP neurons, so it seems to us that MLP neurons may also have the superposition issue. Given this, we think that SAEs are simply not very good at solving the superposition problem (as opposed to, superposition is not a problem at all!), so it makes more sense to try and study models directly (via e.g. MLP neurons) for many practical applications.

---

> > ### Author Rebuttal · Reviewer_VC5S · 2026-04-03
> >
> > Thank you for the clarification. I am a bit torn between "accept" and "weak accept". It would have been easier for me if RelP was not discovered earlier by Jafari et al. (2025). (The authors claim that they discovered RelP at the same time, but the first version of (Jafari et al. 2025) was published already on 28. Aug 2025, several months before the ICML'26 submission deadline.)
> > In the end, I have decided to follow my gut feeling which very slightly leans towards "accept".

---

### Decision · Program_Chairs · 2026-04-30

**Decision:**

Accept (spotlight)

**Comment:**

This paper challenges the common assumption that sparse autoencoders are necessary for circuit tracing, showing that MLP activations can serve as a comparably sparse and effective feature basis. Reviewers agree that this is a novel and potentially impactful finding, supported by the useful attribution method (RelP) and related experiments. Concerns focused on limited datasets, missing baselines, and presentation clarity. In the rebuttal, the authors addressed these issues, including additional experiments and clarifications, which reviewers found satisfactory.


Overall Recommendation: Accept.